



# Mesoscale simulations of tropical cyclone Enawo (March 2017) and its impact on TTL water vapor.

Damien Héron[1], Stephanie Evan[1], Joris Pianezze[1,2], Thibaut Dauhut[3], Jerome Brioude[1], Karen Rosenlof [4], Vincent Noel[5], Soline Bielli[1], Christelle Barthe[1] and Jean-Pierre Cammas[1,6]

[1]Laboratoire de l'Atmosphère et des Cyclones, UMR8105 (Université de La Réunion, CNRS, Météo-France)
[2]Mercator Ocean, Ramonville Saint-Agne, France
[3]Max Planck Institute for Meteorology, Hamburg, Germany
[4]Chemical Sciences Laboratory, Earth System Research Laboratory, NOAA, Boulder, 80305, CO, USA
[5] Laboratoire d'Aérologie, CNRS/UPS, Observatoire Midi-Pyrénées, 14 avenue Edouard Belin, Toulouse, France
[6]Observatoire des Sciences de l'Univers de La Réunion, UMS3365 (CNRS, Université de La Réunion, Météo-France), Saint-Denis de la Réunion, France

*Correspondence to*: Stephanie Evan (stephanie.evan@univ-reunion.fr) and Damien Héron (damien.heron@univ-reunion.fr)

**Abstract.** In early March 2017, tropical cyclone (TC) Enawo formed north of Réunion Island and moved westward toward Madagascar. Enawo evolved from a tropical depression on 2 March to an intense TC on 6 March. This study explores the water vapor transport into the tropical tropopause layer (TTL) throughout TC Enawo's development. High-resolution (2km) mesoscale simulations using the Meso-NH model were performed to cover TC Enawo's lifecycle over the ocean for the period 2-7 March 2017. The simulated convective cloud field agrees with geostationary satellite infrared observations. Compared to the Global Precipitation Measurements (GPM) and Cloud-Aerosol Lidar and Infrared Pathfinder Satellite Observation (CALIPSO) satellite observations, the simulation seems to reproduce well both location and amplitude of the observed precipitation. Simulated and observed ice water content have similar ranges in the upper troposphere but simulated ice above the tropopause is overestimated by a factor 10. Balloon-borne measurements of water vapor, temperature and horizontal winds are also used to validate the Meso-NH simulations in the upper-troposphere and TTL regions. The simulations reveal that the maximum water vapor transport into the TTL occurred on 4 March, when deep (cold) convective clouds were observed. As a result, the lower stratospheric water vapor is increased by ~50% when compared to pre-storm conditions. An increase of ~2ppmv in water vapor mixing ratio was simulated in the lower stratosphere within a 700-km region surrounding Enawo's center. Our simulation of TC Enawo suggests that TCs over the Southwest Indian Ocean (0-30°S, 30-90°E) could produce a moistening of 0.4 ppmv. We extended our results to the global tropics (30°S-30°N) using the estimates from published work (Allison et al., 2018; Preston et al., 2019) and by calculating statistics on TC numbers and durations using the International Best Track Archive for Climate Stewardship (IBTrACS) dataset. We estimated a global impact of TC induced tropical lower stratospheric moistening of 0.3 to 0.5 ppmv. Our results suggest that TCs may play an important role in the moistening of the TTL/lower stratosphere via direct injection of ice particles and subsequent sublimation.

## 1. Introduction

Water vapor in the low stratosphere regulates up to 10% of the greenhouse effect at the earth's surface (Solomon et al., 2010). The stratospheric water vapor concentration is primarily affected by the freeze-drying at the tropical tropopause (Brewer, 1949) and the production of water vapor by methane oxidation (le Texier et al., 1988). It has



been established that freeze-drying process at the cold point temperature of the tropical tropopause is the main

driver of the annual and interannual variability in the entry value of stratospheric water vapor (Mote et al., 1996; Randel et al., 2004). On a long-term and global scale, air enters the stratosphere through the tropical tropopause (Brewer, 1949). This air regulates the radiative and chemical balance of the global Upper Troposphere Lower Stratosphere by carrying trace species with it.  In the tropics, between 14 km and 19 km in altitude, the tropical tropopause layer (TTL), is a transition layer between two different dynamical regimes from rapid vertical transport

in the troposphere associated with moist tropical convection to slow ascent in the stratosphere due to the Brewer Dobson circulation. The TTL is often referred to as the "gateway to the stratosphere" for water vapor and other chemical constituents (Fueglistaler et al., 2009). Processes occurring in the TTL control the hydration of the stratosphere (Jensen et al., 2013). The overall contribution due to transport of air by passing the cold trap is not well quantified, but it may contribute during extreme events (e.g. the strong 2016 El Niño; Avery et al., 2017).

Very deep convection may overshoot the tropical tropopause (~17km), injecting water vapor and ice crystals directly into the stratosphere (Avery et al., 2017; Corti et al., 2008). In addition, overshooting convection plays a critical role in transporting short-lived chemical compounds from near the surface to the lower stratosphere, which is hard to achieve via a large-scale slow ascent (Bergman et al., 2012; Shepherd, 2008).

Dessler et al. (2016) hypothesize that injection of ice into the lower stratosphere could account for 20 to 50% of

the 1 ppmv increase in lower stratospheric water vapor expected in a warming climate. As the role of tropical convection on the Upper Troposphere/Lower Stratosphere (UTLS) composition is still under discussion, observational and modeling studies are needed to further assess the net effect of convection on UTLS water vapor and other short-lived chemical compounds.

Deep convection and in particular overshooting convection have a direct impact on lower stratospheric water

vapor. The net effect (hydration or dehydration) will depend on the depth of the injection, the size of the injected ice particles, and the conditions (i.e. relative humidity with respect to ice RHi) of the environment into which the overshoots inject ice (Dauhut et al., 2018; Jensen et al., 2007; Schoeberl et al., 2018; Ueyama et al., 2018) In sub-saturated TTL air, condensed ice is not removed quickly enough to produce net dehydration (Jensen et al., 2007). Deep convection associated with tropical cyclones can play an important role in the transport of chemical

constituents from the troposphere to lower stratosphere (Ray and Rosenlof, 2007; Zhan and Wang, 2012).

From 23 years of infrared brightness temperature (IR-BT) measurement, Romps and Kuang, (2009) showed that a large fraction of overshooting tropical convection occurs within tropical cyclones (15% of all the convection that overshoot the tropopause). Using Lagrangian trajectories and airborne measurements, Vogel et al. (2014) showed that the combination of rapid uplift by a typhoon and eastward eddy shedding from the Asian monsoon

anticyclone is a fast transport pathway that carries boundary pollutants from Southeast Asia/West Pacific within approximately 5 weeks to the lowermost stratosphere in northern Europe. Analyses of balloon-borne sensors launched from Lhasa, China, in August 2013 show that nearly half of the measured ozone profiles in the upper troposphere were influenced by tropical cyclones occurring over the western Pacific (Li et al., 2017).

Several studies based on model results have also demonstrated that deep convection can efficiently transport water

vapor and pollutants to the UTLS (Hassim and Lane, 2010; Jensen et al., 2007; Marécal et al., 2006).  Recently, Allison et al. (2018) used high-resolution (1.33km) simulations with the Weather Research and Forecasting (WRF) model to investigate the water vapor transport to the UTLS within TC Ingrid (a category 1 storm on the



Saffir Simpson Scale) in the Gulf of Mexico. They identified dehydration between 14.5 and 17.5 km due to ice sedimentation and hydration between ~17.5 and 21 km due to the sublimation of ice crystals.

In addition, Preston et al. (2019) investigated transport of O3/CO/water vapor by typhoon Mireille (a category 3 storm) over the Western North Pacific using the WRF model with chemistry (WRF-Chem). Their high-resolution (3km) simulations showed positive vertical fluxes throughout the troposphere and the tropopause, which increased the CO/water vapor concentration in the UTLS region. Despite having different intensities (category 1 versus category 3 storm), overshooting tops were identified for both systems that transported large quantities of water

vapor to the UTLS. Both studies also agree in terms of estimation of net water vapor flux within 50%.

The current study focuses on an intense TC that occurred over the SWIO in March 2017. (Tao and Jiang, 2013) For the period 1979-2008, an average of ~90 TCs having maximum sustained winds ≥ 63 km/h occurred each year globally (cf. Table 2.5 WMO 2017). The most active TC basin is the Western North Pacific (WNP, ~30% of the global total), followed by the Eastern North Pacific (ENP, ~19% of the global total), the Southwest Indian

Ocean (SWIO, ~15% of the global total) and the North Atlantic (NA, ~13% of the global total). The SWIO has been poorly studied so far despite having tropical-cyclone activity that is comparable to that of the NA. Using 11-year Tropical Rainfall Measuring Mission (TRMM) precipitating hydrometeor satellite observations, Tao and Jiang, (2013) identified overshooting tops in tropical cyclones (above 14 km) and showed that the South Indian Ocean is the second basin after the Northwest Pacific in terms of total number of overshooting tops (cf. Table 2

of Tao and Jiang, 2013).

The event analyzed is the intense TC Enawo (category 3 on the US Saffir Simpson Scale), that made landfall in northeastern Madagascar on 7 March 2017, killing more than 80 people and causing extensive damage. In a previous study, Evan et al. (2020) investigated the effect of deep convection on the TTL composition over the SWIO during austral summer (December-March). They focused on the origin of convective signatures in two

balloon-borne water vapor profiles observed in the vicinity of tropical storms in January 2016 (Tropical Storm Corentin) and March 2017 (TC Enawo). The balloon-borne measurements were made in coordination with lidar observation at the Maïdo Observatory on Réunion Island (21°S, 55°). Using lagrangian backtrajectories, convective activity in both tropical storms was shown to produce significant hydration in the upper troposphere (UT). In contrast, no water vapor anomaly was found near or above the tropopause region on 3 March 2017 over

Réunion Island as the tropopause region was not downwind of TC Enawo. In addition, the balloon was launched at a distance of ~1000 km from Enawo when the storm was still intensifying. Results from Allison et al. (2018) and Preston et al. (2019) suggest that overshooting convection and subsequent water vapor transport to the lower stratosphere (LS) mostly occurs in the eyewall region of TCs. Here, we extend the Evan et al. (2020) analysis of TC Enawo using a mesoscale model to quantify the impact of TC Enawo on UTLS water vapor.

With a background ranging from 2 ppmv to 6 ppmv (Fueglistaler et al., 2009; Rosenlof et al., 2001), the water vapor content in the TTL is very sensitive to small variations. Most GCMs with coarse horizontal and vertical grid spacings underestimate convective transport and its effect on UTLS water vapor. High-resolution modeling allows a better representation of deep convection within TCs and is therefore useful to understand the relative roles of vertical transport of and ice microphysics on UTLS water vapor (Allison et al., 2018; Chaboureau et al.,

2007; Dauhut et al., 2015; Frey et al., 2015; Mrowiec et al., 2012; Ravindra Babu et al., 2015). The main goal of this study is to further our understanding of the impact of TC on the TTL by simulating the TC Enawo over the SWIO basin, for the period 2-7 March 2017 when the storm was intensifying from tropical storm to intense tropical





cyclone. over the SWIO. We take advantage of the Meso-NH model that has been previously developed for the simulations of TCs in the SWIO (Hoarau et al., 2018; Pianezze et al., 2018). This study is also part of the
CONCIRTO (CONvection CIRrus over the Tropical indian Ocean) project that aims to further our knowledge on how deep convection and cirrus clouds affect the TTL over the SWIO.

The present paper is organized as follows. Section 2 describes the Meso-NH model setup as well as balloon-borne in situ measurement and satellite observations used to evaluate the model representation of TC Enawo. Section 3 presents an overview of the evolution of TC Enawo in both observations and simulations. Section 4 investigates
simulated water and ice transport to the low stratosphere. The global impact of TCs to the tropical lower stratospheric water vapor is assessed in Section 5. The model results are discussed in Section 6. Finally, Section 7 contains the conclusions and a summary of our study.

## 2. Observations and Mesoscale model

### 2.1 Satellite observations

METEOSAT 8 is a geostationary satellite located at 41.1°E that monitors for the Indian Ocean since March 2017. The Brightness Temperatures (BT) of the infrared channel at 10.8 µm are used to identify deep convective clouds within TC Enawo and validate their representation in the Mesoscale simulations. METEOSAT 8 data used in this study are provided by National Oceanic and Atmospheric Administration National Centers for Environmental Prediction (NCEP) and correspond to globally merged (60°S-60°N) infrared BT from various geostationary
satellites (John Janowiak, 2017). They have a temporal resolution of 30 minutes and a horizontal resolution of 4 km.

The Global Precipitation Measurement Integrated Multi-satellitE Retrievals (GPM-IMERG) product from the National Aeronautics and Space Administration (NASA) is used to evaluate simulated rainfall. It uses an algorithm that merges precipitation radar, microwave precipitation estimates, microwave-calibrated infrared, and
rain gauge analyses at a spatial resolution of 0.1∘ over the latitudinal belt 60°N–60°S (Huffman et al., 2018). The GPM-IMERG product has a temporal resolution of 30 minutes.

Measurements from the Cloud and Aerosol Lidar with Orthogonal Polarization (CALIOP) lidar onboard the Cloud-Aerosol Lidar and Infrared Pathfinder Satellite Observation (CALIPSO) satellite are used to validate the distribution of simulated ice clouds in the UTLS. Original CALIOP observations correspond to backscatter
measurements at 532 nm and 1064 nm since June 2006 (Level 1 data). CALIOP Level 2 data (version V4.10) contain extinction and Ice Water Content (IWC) profiles at 60-m vertical and 5-km horizontal resolutions, which are retrieved from the 532-nm extinction coefficient (Avery et al., 2012). Between March 2 and 7, there were 9 CALIPSO overpasses over TC Enawo. However, only the overpass on 5 March at 21:30 UTC was over TC Enawo's eye region (cf. Figure 7).

### 150 2.2 Balloon-borne observations.

A balloon launch was specifically planned using a Lagrangian model and geostationary infrared images to sample the convective outflow from TC Enawo on 3 March 2017 at 18 UTC. The balloon sonde payload consisted of the Cryogenic Frostpoint Hygrometer (CFH) as well as the Intermet iMet-1-RSB and Meteomodem M10 meteorological radiosondes. A detailed description of the balloon measurements is provided in Evan et al. (2020).


The Cryogenic Frost Point Hygrometer (CFH) is an in-situ instrument that measures the water vapor mixing ratio
profile from the surface to the stratosphere (~28km).

The CFH was developed to provide highly accurate water vapor measurements in the TTL and stratosphere where
the water vapor mixing ratios are extremely low (~2 ppmv). CFH mixing ratio measurement uncertainty ranges
from 5% in the tropical lower troposphere to less than 10% in the stratosphere (Vömel et al., 2007) ; a recent study

shows that the uncertainty in the stratosphere can be as low as 2-3% (Vömel et al., 2016).   The M10 radiosonde
provides measurements of Relative Humidity (RH), temperature, pressure, vertical velocity, wind speed and
direction from which zonal/meridional winds are derived.

### 2.3 Meso-NH model

Simulations of TC Enawo were performed with the Mesoscale, anelastic, and nonhydrostatic model Meso-NH

version 5.3 (Lac et al., 2018). Recent Meso-NH simulations have been used successfully to study tropical
storms/cyclones over the SWIO (Barbary et al., 2019; Hoarau et al., 2018; Lac et al., 2018; Pianezze et al., 2018).
Meso-NH has also been used to study overshooting convection and stratospheric hydration (Chaboureau et al.,
2007; Dauhut et al., 2015).

Different simulations (Table 1), were performed with the Meso-NH model. The simulations have vertical grid

spacings less than 100 m in the boundary layer and 300m up to 30km (140 verticals levels), with a damping layer
in the uppermost 25 km and a model top at 1hPa. All simulations are initialized and forced at the boundaries with
6-hourly analyses from the operational European Centre for Medium Range Weather Forecasts - Integrated
Forecast System (ECMWF-IFS) with a grid spacing of ~9km and 137 vertical levels. Simulations were run from
2 March 2017, 00UTC to 7 March, 00UTC to encompass Enawo's evolution from tropical depression to very

intense TC just before landfall over Madagascar. The turbulent scheme is based on Cuxart et al. (2000). We use
a single-moment bulk mixed-phase cloud parameterization ICE3 (Lac et al., 2018; Pinty and Jabouille 1998). It
solves the microphysics of five hydrometeors which are three precipitable species (snow, rain, and graupel) and
two non-precipitating hydrometeors (cloud water, and cloud ice). The longwave radiative scheme used in Meso-
NH is the rapid radiation transfer model (RRTM; Mlawer et al., 1997). The shortwave radiative scheme is based

on Foucart and Bonnel (1980). The subgrid shallow convection is based on the eddy-diffusivity mass flux and
Kain-Fritsch approach of entrainment and detrainment in cumulus clouds (Pergaud et al., 2009). Meso-NH is
coupled with the surface model SURFEX (Surface Externalisée, Masson et al., 2013) and uses COARE-3
parameterisation for ocean-atmosphere fluxes (Fairall et al., 2003). The first model level is at 10m.

Initial simulations were run to test the model's sensitivity to different parameterizations on a domain A covering

46°E-69°E in longitude, 32°S-8°S in latitude at a horizontal grid-spacing of 10 km (Simulation S1). Tests on
domain limits have been performed to identify the sensitivity to the eastern boundary. Then, five simulations were
performed (Table 1) to test the horizontal resolution, nested domains and sensitivity to SST. Simulations were run
in domain A with a horizontal grid-spacing of 10km and 5km (Simulations S1 and S2). The Kain Fritsch cumulus
parameterization was used to simulate deep convection in S1 and S2. A second domain, called domain B, centered

over a region that encompasses Enawo's life cycle was designed (Figure 1) for simulations with a horizontal grid-
spacing of 2km (Simulations S3, S4 and S5). Deep convection was explicitly resolved at that resolution.
Simulation S3 had nested domains having grid spacings of 10 (domain A) and 2km (domain B). It was found that
using one-way grid-nesting did not improve the model's representation of Enawo, so higher resolution simulations



at 2km were run without nesting (Simulations S4, S5). The sensitivity to the Sea-Surface-Temperature (SST) was
also tested. In simulations S1 to S4, the Sea-Surface-Temperature (SST) data from ECMWF were used and
updated every 6 hours. Simulation S5 is coupled with the oceanic model CROCO (Coastal and Regional Ocean
COmmunity model, http://www.croco-ocean.org, (Debreu et al., 2012; Penven et al., 2006). Further details on the
coupling are provided in  Voldoire et al. (2017) and Pianezze et al. (2018). The oceanic model has a horizontal
resolution of 2km and 32 vertical levels from the surface to the ocean bottom (about a depth of 5km below Enawo)
and shares the same domain of simulation as the Meso-NH model.

## 3. Evolution of TC Enawo

### 3.1 TC Enawo (March 2017)

The Madden Julian Oscillation (MJO) was active at the end of February and during the first week of March 2017
with a signal centered over Africa and the Indian Ocean. Favored by the MJO active phase and the arrival of an
equatorial Rossby wave, Enawo initially formed as a tropical disturbance on March 2 with 10-minute maximum
sustained wind speeds ~ 40 km.h$^{-1}$. The Regional Specialized Meteorological Center (RSMC) La Réunion, in
charge of TC's forecasts for the SWIO, named the storm Enawo on 2 March 2017 at 00 UTC. The best track data
used in this study are provided by RSMC La Réunion.

In the SWIO, different stages of TCs are defined using a 10-min maximum sustained as follows:  Tropical
Depression (TD, < 17.4m s-1), Moderate Tropical Storm (MTS, ≥ 17.4 m.s$^{-1}$), Severe Tropical Storm (STS, ≥
24.6 m.s$^{-1}$), Tropical Cyclone (TC, ≥ 32.9 m.s$^{-1}$ = Category 1 on the US Saffir-Simpson Scale), Intense TC (ITC,
≥ 43.6 m.s$^{-1}$, Categories 2-3 on the US Saffir-Simpson Scale) and Very Intense TC (VITC, ≥ 59.6 m.s$^{-1}$, Categories
4-5 on the US Saffir-Simpson Scale).

Between 2 and 4 March, Enawo intensified slowly from TD to TS because of a strong east-southeast vertical wind
shear. On 4 March, Enawo stalled over the ocean while intensifying. Very cold cloud tops (<-90°C) were observed
at that time. Enawo evolved from TS to TC on 5 March at 06UTC with an eyewall with a poorly defined structure.
Later, the intensification speed was slowed by an Eyewall Replacement Cycle (ERC), observed on satellite
microwave images. An ERC occurs when the pressure forces at the center of the TC push inward the eyewall. At
some point, the eyewall collapses and a new eyewall forms afterward. Because of a decrease in vertical wind
shear, Enawo further intensified to reach the ITC stage on 6 March at 12UTC (Category 2-3 on the Saffir-Simpson
scale), with 10-minute maximum sustained winds of 46.3 m s-1 (90kts). The 10-minute maximum sustained winds
(VMAX) increased by 23 m.s$^{-1}$ (45 kts) in 24 hours, larger than the top 5% percentile of rapid intensification over
the SWIO basin (Leroux et al., 2018). Enawo reached peak intensity on 7 March at 06 UTC, with ten-minute
maximum sustained winds at 56.5 m.s$^{-1}$ (110 kts) and the central pressure at 932 hPa. TC Enawo reached
Madagascar's northeastern coast on March 7 at around 09:30 UTC and was the third strongest tropical cyclone on
record to strike the island. It was also the strongest TC of the Southern Hemisphere for the 2016/2017 TC season.
The RSMC La Réunion provides 6-hourly best-track data that include the location of the storm center, minima of
mean sea level pressure (MSLP) and 10-min maxima sustained wind. Root mean square errors (RMSE) between
the Best Track and the simulations were calculated for track (in km) and MSLP (in hPa) following the approach
of Chandrasekar and Balaji (2012) and Allison (2018). The RMSE for the track and MSLP for each simulation
are presented in Table 1.  Based on the results of MSLP and RMSE, simulation S4 does a better job in representing



the intensification phase and the trajectory of Enawo. The comparison of S4 to Enawo's best track, precipitation and balloon-borne measurements are presented in the subsequent sections. Simulation S4 will be used in section 4 to investigate the vertical structure of the TTL and water vapor/ice transport to the lower stratosphere.

### 3.2 Trajectory/intensification

Figure 1 compares the simulated trajectory of simulation S4 to the best-track data from March 2 at 06UTC (top right) and March 7 at 00 UTC (bottom left). The trajectory in the simulations is defined using a minimum of Mean Sea Level Pressure (MSPP). The simulated trajectory is to the south of the one provided in the best-track data but with a RMSE of 97±54km. After March 4, the simulated trajectory is even more to the south when compared to the best track. However, the westward propagation of the tropical cyclone is relatively well represented with comparable propagation speed. On March 4, the simulation was able to show that Enawo stalled over the ocean near 13.6°S/56.5°E. In the rest of the study, we use the minimum MSLP to define the storm center.

Figure 2 compares Enawo's simulated intensity (MSLP) with the corresponding RSMC La Réunion best-track data. Enawo's intensity is relatively well represented in the simulation in the first 60 hours of the run (2,3 and 4 March). The simulation has a RMSE for MSLP less than 6hPa. Since no nudging was applied, it is remarkable that the Meso-NH simulations can represent Enawo's initial development from tropical depression on 2 March, 00UTC to moderate tropical storm on 3 March, 18UTC. This is most likely due to the high-resolution operational ECMWF analyses used for initialization and the fact that the initial storm structure was present in the ECMWF analysis on 2 March 00UTC. When initialized a few days before, the Meso-NH did not generate the initial tropical depression that would become TC Enawo.

On 5 March 00UTC, Enawo stalled over the ocean after the ERC. The ERC was not reproduced by the simulation and the simulation started to diverge compared to Enawo's MSLP. An ERC is particularly difficult to simulate and a resolution less than 500 m is necessary to resolve this process (S. Bielli, personal communication). Between 60h and 96h, the intensification is slower in the simulation as indicated by the slower decrease in the simulated MSLP. After 96h (6 March, 00 UTC), the rapid intensification when Enawo evolves from TC to intense TC is shown in the best-track data with a drop of 30 hPa in the MSLP over 12 h. The model is not able to capture this rapid intensification but instead simulates a steady intensification phase. As a consequence, the error in the MSLP increases to 20 hPa at the end of the simulation.

### 3.3 Precipitation

Figure 3 displays a comparison of GPM observed rainfall (left) to the one simulated in the S4 simulation (right). For this comparison, simulated precipitation was interpolated to the GPM grid (0.1°x0.1°) and integrated over the period 2 March, 06 UTC to 7 March, 00 UTC. Overall, the simulated total rainfall agrees with observations and Meso-NH captures the north-south asymmetry.

Between March 4 and March 5, heavy rain (in red on Figure 5) is observed in the eyewall. Precipitation for this period tends to be overestimated in the S4 simulation. Precipitation within the secondary rainbands are consistent with those observed, especially south of Enawo's trajectory. The model tends to produce lighter precipitation over a larger region. In addition, simulated precipitation over the eastern coast of Madagascar is not observed in the GPM data.





### 3.4 Comparison to balloon-borne measurements.

We further compare the Meso-NH model results to balloon-borne measurements at the Maïdo Observatory on 3 March 2017 at 18 UTC. At the time of the balloon launch at the Observatory, Enawo was a tropical storm located near 13° south latitude and 56.42° east longitude, about 900 km northwest of Réunion Island. For this comparison, the model fields of water vapor mixing ratio, temperature, zonal and meridional winds are interpolated onto the balloon's location and put on a regular 200-m vertical grid. Comparisons between the simulated profiles and

CFH/M10 radiosondes are shown on Figure 4. The CFH water vapor profile has fine-scale structures at 6 and 10km which are not reproduced in the simulation (Figure 4, right).

The detrainment of water vapor at ~13 km is resolved in the simulation but remains lower than the observed one. In the troposphere between 2 and 14 km in altitude, the simulation has a wet bias of +450 ppmv, resulting from a higher moisture transport with an explicit representation of convection at a 2 km grid-spacing. In the TTL (14-

18km in altitude), the mean water vapor difference at the location of the Maïdo Observatory is about -1ppmv in S4.

The simulated temperature profile is colder than the M10 temperature profile in the troposphere between 2 and 14 km with a mean bias of -1 K. A cold bias of -2 K is also observed in TTL (14-18 km). However, both the model and the observation indicate a CPT at ~16 km.

On Figure 4, the simulated and observed profiles of zonal and meridional wind speed are in good agreement. The peak in meridional wind at 13km associated with Enawo's convective outflow in the UT is captured by the model.

### 3.5 Infrared brightness temperature

Figure 5 compares simulated (top panel) and observed (bottom panel) infrared Brightness Temperatures (BT) at different stages of Enawo's lifecycle. The simulated infrared BT is estimated using the RTTOV-v11 radiative

model RTTOV-v11 (Saunders et al., 2018; Senf and Deneke, 2017). Infrared BTs are shown on 3 March, 18 UTC (time of the balloon launch, left panels), 4 March 12 UTC (middle panels) and 5 March 21:30 UTC (right panels). March 4 and 5 correspond to days of intensifying convection within Enawo that became a TC on 5 March at 12UTC. The simulated horizontal structure of Enawo is relatively well represented when compared to the METEOSAT 8 satellite images.

The development of convective clouds is slightly delayed by a few hours in the simulation and the eyewall region is not as well defined as in the observation. On 3 March 18 UTC, the simulation overestimates convection over the eastern part of the domain. On 4 March 12 UTC, Enawo's deepest (coldest infrared BT) convective clouds are located north of 15°S. Simulated convective clouds are 2°C colder than the observed ones which suggest higher convective clouds than observed. In the simulation, the coldest infrared brightness temperatures range from -85°C

(March 2) to -92°C (March 4&6) while the METEOSAT 8 infrared BT range from -80.5°C to -90°C.

On March 5 at 21:30 UTC, the spiral rainbands are not as well defined in the simulation as clouds are more scattered than in the observation. Simulated convective clouds are overestimated in the northwest quadrant while they are underestimated in the southeast quadrant of the storm. Cold infrared BTs are observed in the simulated TC rainband which suggest the existence of updraft in the simulation. There was a CALIPSO overpass over the

eyewall on 5 March, 2130 UTC (orange lines on Figure 7) and the distribution of simulated ice is presented in the subsequent section.



## 4. Results

### 4.1 Vertical structure in the TTL.

Figure 6 represents an along-track cross-section (latitude versus altitude) comparing IWC profiles from the CALIOP observations to the Meso-NH simulation. The simulated IWC variable includes both cloud ice and precipitating snow which are prognostic variables of the ICE 3 microphysics scheme. The cross-section of IWC is shown for CALIPSO's orbit track on 5 March 21:30 UTC, crossing Enawo's eye region. At that time, Enawo was a severe Tropical Storm and the position of the simulated storm agrees with the geostationary satellite image valid at that time. On Figure 6, the mean cold point tropopause in the simulation is at 16.5 km and is indicated in 315 black on both panels.

The CALIOP lidar cannot penetrate deep convective clouds, therefore observed IWC is not shown below ~10 km. From the distribution of IWC on Figure 8, we can see that the model tends to produce less ice in the convective core below ~15km in altitude, and more ice above. On the contrary, less ice is produced in the upper troposphere in Enawo's outer regions (south of 18.5°S and north of 11°S). In the upper troposphere, IWC values range from 320 $10^{-4}$ to $10^{-2}$ g.m$^{-3}$ in both the observation and the simulation. Maximum values of IWC are observed in the eyewall: ~0.01 to 0.05 g.m$^{-3}$ in the observation versus ~0.05 to 0.1 g m$^{-3}$ in the simulation. Simulated IWC is a factor 10 larger than CALIPSO IWC above the CPT. The location of the convective center is fairly well reproduced by the simulation (13°S,55°E in S4 versus 14°S,54.7°E in the CALIPSO observation). The simulation does show ice above the CPT between 11 and 15°S (IWC ranging from $10^{-5}$ to $10^{-4}$ g.m$^{-3}$). However, CALIOP only shows ice 325 above the CPT in the eye region (according to the best-track data the storm center was located at 13.96°S, 55.02°E on 5 March, 18 UTC).

Average mass ($\varphi_{mass}$), ice ($\varphi_{ice}$) and water vapor ($\varphi_{vapor}$) flux density (kg.m$^{-2}$.s$^{-1}$) to the low stratosphere (~18 km) can be computed as a function of time and distance from the TC center using the equations:


$$\varphi_{mass} = \frac{\left(\sum_{r_i < r_k}^{r \leq r_{k+1}} \rho(r_i) w(r_i)\right)}{N_k} \qquad (1a)$$

$$\varphi_{ice} = \frac{\left(\sum_{r_i < r_k}^{r \leq r_{k+1}} \rho(r_i) w(r_i) q_{ice}(r_i)\right)}{N_k} \qquad (1b)$$

$$\varphi_{vapor} = \frac{\left(\sum_{r_i < r_k}^{r \leq r_{k+1}} \rho(r_i) w(r_i) q_{vapor}(r_i)\right)}{N_k} \qquad (1c)$$

$r_i$ corresponds to a model grid point, $\rho$ is the air density (kg.m$^{-3}$), $q_{ice}$ is the ice mixing ratio (kg.kg$^{-1}$), $q_{vapor}$ is the water vapor mixing ratio (kg.kg$^{-1}$) and w is the vertical velocity (m.s$^{-1}$). Figure 7 illustrates the calculation of $\varphi_{mass}$ and can be applied to $\varphi_{ice}$ and $\varphi_{vapor}$ by adding the $q_{ice}$/ $q_{vapor}$ term. We sum the $\rho \times w$ term at individual grid point over ring circle regions around the TC center (red symbol on Figure 7) defined by radiuses ($r_k$) varying from 20 to 700 km by 20 km increment (ie., $r_{k+1}$- $r_k$ = 20 km) and divide by the number $N_k$ of grid points in the ring circle 340 region to define average $\varphi_{mass}$, $\varphi_{ice}$, $\varphi_{vapor}$ (kg.m$^{-2}$.s$^{-1}$). Some studies have proposed to calculate upward flux using only grid points containing upward vertical motion (Chaboureau et al., 2007; Dauhut et al., 2016; Mrowiec et al., 2012). Wei, (1987) has estimated mass transport across the tropopause by looking at the contribution from diabatic processes, the temporal change of the tropopause potential temperature and mass exchange due to the potential temperature gradient along the tropopause (Ravindra Babu et al., 2015; Wei, 1987).



The maximum mass flux density is simulated on 4 and 6 March around 00 UTC within ~100km of the TC center (eye and eyewall regions) with values ranging from 0.5 to $1\times10^{-3}$ kg.m$^{-2}$.s$^{-1}$. Similarly, the maximum water vapor flux density is observed on 4 and 6 March, with values ranging from 1.5 to 2.5 $\times10^{-8}$ kg.m$^{-2}$.s$^{-1}$. Large negative values are simulated on 4 March at 12UTC for both mass and water vapor fluxes density. During the same time period, sporadic transport of ice across 18km is simulated within a 300 km region around Enawo's center. At that

time, Enawo was still forming and thus updraft regions were more scattered within the storm. TC Enawo stalled over the ocean on March 4&5 and intensified (Figures 1&2). An enhancement of ice flux in the eye/eyewall regions (150 km around the TC center) can be observed on those days which is consistent with high (cold) convective clouds observed on Figure 5. After March 5, the ice flux at 18 km decreased although the observed storm reached peak intensity on 7 March. Comparison of the ice and water vapor fluxes on Figure 8, clearly

indicates that positive water vapor flux density into the stratosphere doesn't necessary mean direct injection of ice, and vice versa.

**4.2 Overshooting convection on 4 March.**

   As indicated on Figure 8, positive ice flux to the LS was observed on 4 March, when the storm was intensifying over the ocean. In this section, we show an example of an overshooting convective event that occurred on 4 March

between 11:55 UTC and 12:45 UTC (Figure 9). Figure 9 depicts north to south cross sections (at longitude 56.09°E) through the simulated overshooting cloud on 4 March. Simulated higher vertical velocity near 13.4°S is coherent with the location of the storm center in the best-track data (13.65°S, 56.85°E on 4 March, 12UTC).

   At 11:55 UTC, vertical winds in the lower TTL troposphere (14-16.5 km) are relatively weak. The 380 K isentropic surface (black lines on Figure 9) is deformed, possibly due to eddies or convectively generated gravity

waves. A dry layer can be observed below 380 K, which is linked with the cold point tropopause. Higher values of water vapor mixing ratio ~20 ppmv above the tropopause could be due to a precedent overshooting event within Enawo.

   At 12:05 UTC, a stronger updraft develops near 13.5°S which further deforms the 380 K isentropic surface. Direct ice injection can be seen between 17 km and 19 km (ice mixing ratio of 10 ppmv to 1000 ppmv at 18 km) and

dehydration below the tropopause between 13.4°S to 13.6°S becomes more important.

   At 12:15 UTC, stronger upward motion and tropopause's deformation are still observed. The water vapor mixing ratio above 380K increases from 15 ppmv to more than 45 ppmv.

   10 minutes later, at 1225 UTC ice mixing ratio values decrease by a factor of 100 in the lower stratosphere (17-19km, 13.4°S-13.6°S) while water vapor mixing ratios increase by a factor 4 compared to the initial condition (40 versus 10 ppmv initially).

375    This suggests that ice was transported in the lower stratosphere and sublimated in a subsaturated environment thereby hydrating the LS (the full processes are described at high temporal resolution in Dauhut et al., 2018). The 380 K potential temperature surface between 13.4°S and 13.6°S is smoother and returns to its initial altitude (17km). The deformation of the 380 K surface between 12:05 UTC to 12:45 UTC suggests first strong deformation by the overshoot and then horizontal vertical propagation of high-frequency

380    gravity waves triggered by deep convection. At 12:35 UTC, ice mixing ratio contours become more similar to their initial values in the LS. Ice, which previously sublimated, is diluted in the LS. At 12:45 UTC, vertical motion is very weak in the lower TTL and the 380 K isentropic surface has returned to its initial 11:55 UTC altitude.





During the overshooting event on 4 March, the average LS water vapor mixing ratio increased from 6.1 to 7.4 ppmv around the TC center. It suggests transport of ice/water vapor from the troposphere. Ice sublimation in a sub-saturated LS leads to moistening. These results are further discussed in the discussion section.

### 4.2 Water vapor in TTL

We estimated the temporal change in TTL water vapor mixing ratio between the beginning and end of the Meso-NH simulations. Water vapor profiles were averaged over a 500 km region surrounding Enawo on 2 March 06 UTC (6 hours after the start of the simulation) and 7 March 00 UTC (end of the simulation). The average water vapor mixing ratio difference between these two dates is shown on Figure 10 for the Meso-NH simulation. A minimum of water vapor hydration can be seen at 16.5 km, near the tropical tropopause. A weak dehydration of -0.1 ppmv can be seen near the tropopause at 16.5 km. Using CFH and MLS water vapor profiles, Evan et al. (2020) estimated dehydration ranging from -0.4 to -0.1 ppmv at 100 hPa from a monthly mean MLS climatology (Figure 9 of Evan et al., 2020). Between 17 to 19km an increase in water vapor mixing ratio up to 2 ppmv can be seen. Allison et al., 2018 reported hydration between 17.5 and 21 km associated with Hurricane Ingrid (Category 1) over the North Atlantic, with a maximum of 2.6 ppmv. Ice crystals were transported to the LS by deep convection, sublimated and produced hydration.

Figure 11 shows time versus distance from Enawo's center simulated lower stratospheric water vapor mixing ratio, averaged over the 17-19 km altitude range, from 2 March, 06 UTC to 6 March, 18 UTC. Enhanced mixing ratio of water vapor, ~5 to 6 ppmv compared to a background mixing ratio of 3.3 ppmv, is observed first on March 3 at 00UTC near Enawo's center. High clouds were indeed observed on March 3, but as shown on Figure 5, the eyewall region in the simulation is not as well defined as in the observation and therefore this hydration event may be not well reproduced by the model. The enhanced mixing ratio has a slant wise propagation along time, with the enhanced mixing ratio propagating about 300 km away from Enawo's center in 12 hours. It shows that the water vapor injected near Enawo's center is advected outward from the storm's center, following the upper divergence wind pattern and Enawo's displacement, in the simulation.

A second hydration event near Enawo's center can be observed after 4 March 12 UTC. It is related to injection of ice in the TTL discussed in section 4.1 and 4.2 and depicted on Figure 8. The water vapor mixing ratio increased up to 7 ppmv in a 100 km radius around Enawo's center. Then, as observed for the injection on 3 March, the water vapor positive anomaly propagates outward Enawo's center, with water vapor mixing ratio of 5 ppmv transported 500 km away from Enawo's storm after 1.5 day of transport. At the end of the simulation, the water vapor mixing ratio background rose from 3.3 ppmv to 4.8 ppmv, an increase of 45%, 500 km away from Enawo's center.

### 5. Global impact of TCs to the tropical lower stratospheric water vapor

We compute a net vertical water vapor flux $F_{mean}$ (t.hr$^{-1}$, where t corresponds to the metric ton) at 18 km which corresponds to the lower-stratosphere using:

$$F_{mean} = \sum_i \rho_i q_{vi} w_i \times A \times 3.6 \quad (2)$$

Where $\rho$ is the density (kg.m$^{-3}$), w the vertical velocity (m.s$^{-1}$) at a grid point, qv the water vapor mixing ratio (kg.kg$^{-1}$) at a grid point, A is the area of a grid cell (2000 x 2000 m$^2$) and the 3.6 factor accounts for the conversion


from $kg.s^{-1}$ to $t.hr^{-1}$. Net vertical water vapor flux was calculated by summing all flux values at individual grid points, regardless of sign, for a region of 700 km surrounding the TC center. It was found that a ~700 km radius best encompassed Enawo's circulation. Allison et al., (2018) used a radius of 300 km to define TC Ingrid in the North Atlantic and Preston et al. (2019) used a radius of 900 km for Typhoon Mireille in the Western North Pacific. We use hourly outputs on 4 and 5 March 2017 (48 hours) to estimate net vertical water vapor flux as it

corresponds to days with larger water vapor fluxes on Figure 8.

Figure 12 displays the evolution of net water vapor flux $F_{mean}$ at 18 km for 4-5 March 2017. The average net water vapor flux of $2.7 \times 10^3$ $t.hr^{-1}$ is positive, indicating that Enawo is a source of vapor for the lower stratosphere. The net water vapor flux varies from -1.3 to $1.7 \times 10^4$ $t.hr^{-1}$ with local peak values on 4 March 06 UTC, 5 March 06 UTC and 20 UTC (Enawo first achieved TC status on 5 March 12 UTC). Chaboureau et al. (2007) found observed

peak water vapor flux of $2.8 \times 10^4$ $t.hr^{-1}$ across the tropical 380 K level (~ 100 hPa) for a case of land-convection over Brazil. Allison et al. (2018) used high-resolution (1.33 km) numerical simulations of TC Ingrid (September 2013) in the North Atlantic to assess vertical water vapor transport to the lower stratosphere. The net water vapor flux at 100 hPa was $2.1 \times 10^3$ $t.h^{-1}$ on 14 September 18 UTC when Ingrid first achieved hurricane status. Over a 43-hour period, net vertical flux values ranged from $-4.6 \times 10^3$ to $1.2 \times 10^4$ $t.hr^{-1}$. They estimated a total mass water

vapor transport of $2 \times 10^5$ t to the 100 hPa level associated with Category 1 TC Ingrid (thus an average net vapor flux of $4.6 \times 10^3$ $t.hr^{-1}$). Similarly, Preston et al. (2019) performed a 3 km explicit convection simulation of Typhoon Mireille (1991) over the Western North Pacific, and found a similar average net water vapor flux to Allison et al. (2018). TC Mireille and TC Enawo have a comparable intensity (maximum sustained winds of 51.4 m s-1 and a central pressure of 925 hPa.) as category 3 TC, but Mireille was a larger system with a 900 km radius.

We can use our estimate of average net water vapor flux to the low stratosphere associated with Enawo to estimate the contribution to stratospheric water vapor from SWIO storms. Using the Best-Track data produced by RSMC La Réunion and focused on the most reliable 18-year period of the geostationary satellite era encompassing the 1999/2000–2016/17 cyclone seasons, we found an average of 4.8 TCs per year formed over the SWIO, in agreement with Leroux et al. (2018). We also estimated an average duration of 74±49 hours (± refers to the

standard deviation) per cyclone. Using an average net water vapor flux of $2.7 \times 103$ $t.hr^{-1}$ from Enawo's high-resolution simulation, we estimate that TCs over the SWIO could transport $9.6 \times 105$ t of water vapor into the lower stratosphere. Figure 10 shows that Enawo moistened the region between ~18 and 20 km. Dauhut et al. (2015) found moistening associated with a Hector thunderstorm between potential temperature levels 380-420 K (~17–18.5 km). Using CALIOP data, Avery et al. (2017) reported that convective ice could be observed up to up to 2

km above the tropopause over the Central Eastern Pacific (Figure 2a, 2b of Avery et al., 2017) due to the strong 2015–2016 El Niño. With a mean air density taken to be 0.1 $kg.m^{-3}$ between 18 and 20 km, TCs over the SWIO (0-30°S, 30-90°E) could produce a moistening of 0.4 ppmv, over a region which represents ~8% of the global tropics (30°S-30°N).

While an individual TC may not contribute much to stratospheric water vapor, collectively all TCs on the globe

may have a larger impact. The global TC transport of water vapor to the stratosphere can be computed as:

$$Global\ TC\ mass\ of\ water\ vapor\ (t) = F_{mean} \times NTC \times DTC \quad (3)$$

Fmean is the mean estimate of net water vapor flux at the tropopause ($t.hr^{-1}$) for an individual TC, NTC is the average global number of TCs and DTC is the mean duration of a TC (in hours). We can estimate that Fmean





ranges from 2.7 to $4.6 \times 10^3$ t.hr$^{-1}$ using the net water vapor flux estimates from Allison et al. (2018), Preston et al. (2019) and our simulation.

We derive NTC and DTC of tropical cyclones from the International Best Track Archive for Climate Stewardship (IBTrACS) (Knapp et al., 2010). IBTrACS is a composite dataset of track estimates for TCs, which combines data from multiple agencies to facilitate analysis. We specifically use data from version 4 for the period of global

satellite coverage 1981-2017. The IBTrACS dataset defines 6 TC basins which are the North Atlantic, the West Pacific, the East Pacific, the South Pacific, the North Indian and the South Indian (Souwestern + Southeastern Indian Ocean). Agencies differ in their definition of maximum winds. Agencies in the United States use a 1-min wind, RSMC New Delhi reports a 3-min wind but most RSMCs use a 10-min wind. To compare our results to previous estimates (WMO 2017 and Schreck et al., 2014), the IBTrACS 10-min maximum sustained winds from

RSMCs Tokyo, Fiji, Wellington, La Réunion, and the Australian Bureau of Meteorology are divided by 0.88 to approximate a 1-min wind (hereafter VMAX). The data from RSMCs Miami and New Delhi are used in their original form. For each of TC basin, we consider only systems that reach the TC stage (ie. VMAX ≥ 64 kt) and compute their mean duration as the period during which VMAX ≥ 64 kt. We also consider VMAX at peak intensity. The results for individual basins and the global tropics are summarized in Table 3. Number of TCs

shown in Table 3 can be compared to Table 2.5 of WMO (2017) and Table 3 of Schreck et al., (2014). The mean global NTC is 45±9 for the period 1981-2010. On average, these systems have a DTC of 78±19 hours. Using equation (3), we can estimate that TCs could contribute annually from 0.3 to 0.5 ppmv to the tropical lower stratospheric water vapor.

Several uncertainties arise for the estimate of global TC transport of water vapor from equation (3). The annual

estimate of global TC water vapor transport has a combined uncertainty of 27% due to the NTCxDTC term in equation (3). But the largest uncertainty in equation (3) arises from the estimate of Fmean which in turn depends on TC structure/intensity and representation of microphysical as well as dynamical processes. It is difficult to assess the uncertainty due to the microphysical processes in the models (thus the uncertainty on water vapor mass mixing ratio and vertical velocity change due to latent heat release). In addition, Dauhut et al. (2017) discuss

uncertainty in Fmean estimate due to an Eulerian versus Lagrangian approach. The Eulerian approach leads to larger estimates that can be attributed to reversible motions like gravity waves, which actually do not transport mass. Mrowiec et al. (2015) compare the Eulerian and isentropic computations of the vertical mass flux for a deep convective event around Darwin. The Eulerian computation based on a positive velocity threshold gives a 50% larger estimate than the isentropic computation. Thus, the Eulerian approach used in Allison et al. (2018), Preston

et al. (2019) and the present study could bias the Fmean flux estimate by up to 50%.

## 6. Discussion

The global TC induced LS water vapor anomalies can be compared to previous studies that have quantified the impact of convection on tropical lower stratospheric water vapor.

Using Lagrangian trajectories driven by 6-hourly interim reanalysis of the ECMWF Forecast (ERA-Interim),

Ueyama et al. (2018) assess the impact of convection on the humidity and clouds in the TTL during boreal summer 2007. They estimated that convection moistens the 100-hPa level by 0.6 ppmv averaged over the 10°S–50°N domain. ~50% of this increase (~0.3 ppmv) is due to the effect of the Asian monsoon (0–40°N, 40–140°E) convection. Nützel et al. (2019) performed a 4-year simulation with the chemistry-transport model Chemical



Lagrangian Model of the Stratosphere (CLaMS) driven by the ERA-Interim data and established that water vapor from the Asian Monsoon region contributes on average at most 0.65 ppmv to the water vapor in the tropical stratosphere at 450 K (~19 km). The Asian monsoon region corresponds to an area roughly 13% of the tropics and is mainly active in July-August. Avery et al. (2017) used CALIOP observations of ice, MLS observations of water vapor and lagrangian trajectories to estimate large anomalies in lower stratospheric water vapor and ice during the 2015–2016 strong El Niño. A 0.9 ppmv tropical lower stratospheric moistening was observed during this event, with 0.5-0.6 ppmv due to anomalously warm Tropical Warm Pool tropopause temperatures during December 2015 and 0.3–0.4 ppmv due to deep convection over the Central Pacific (~11% of the tropics). Therefore, our estimate of global TC tropical lower stratospheric moistening of 0.3 to 0.5 ppmv agrees with previous studies that have considered deep convection on a regional scale.

The Asian Monsoon in the Northern Hemisphere is clearly the most important source region but our analysis suggests that the most intense TCs may contribute to the tropical lower stratospheric water vapor budget, especially in the Southern Hemisphere where there is no similar monsoon circulation. Because climate models and observations predict more intense TC with warming sea surface temperatures (Elsner et al., 2008; Kossin et al., 2020; Sobel et al., 2016), accurate understanding of the stratospheric impact of TC convection is critical in the context of global warming. This is important not only for the lower stratospheric water vapor budget but also for the transport of other chemical species (e.g. ozone, CO, ozone depleting substances) to the UTLS.

There are several limitations to the present study. First, we consider a single TC in the SWIO to assess the net TC water vapor transport to the stratosphere in the SWIO basin. TC Enawo may not be representative of other TCs in other basins as it was an intense TC (the strongest TC of the Southern Hemisphere for the 2016/2017 TC season). Then, large-eddy simulations may be needed to better resolve deep convection and associated convective updrafts, especially in the eyewall region. Heath et al. (2017) used 450 m WRF large-eddy simulations to study deep convection during a real-world case (a case of continental convection over Northern America on 2 September 2013) and concluded that moving from grid-spacing of 1.35 km to 450 m yielded results that better compared with ground-based radar and aircraft observations. Dauhut et al. (2015) found that a grid spacing on the order of 100 m may be necessary for a reliable estimate of hydration in a Meso-NH simulation of a Hector thunderstorm observed on 30 November 2005 over the Tiwi Islands. Large-eddy simulations of TC Enawo would be valuable, especially for convective updrafts in the eyewall region, but would require more computer power.

Finally, we use a single-moment microphysics scheme in our 2-km simulation of TC Enawo. One conclusion from Allison et al. (2016) for TC Ingrid is that double-moment microphysics schemes produce more realistic tropical clouds and precipitation, which are important for the representations of updrafts and transport of ice to the TTL. While our comparison to the GPM satellite observations suggests that the simulation seems to reproduce well both location and amplitude of the observed precipitation, our comparison to CALIOP IWC indicates that ice above the tropopause is most likely overestimated in the simulation. Hoarau et al. (2018) performed a 2-km cloud resolving Meso-NH simulation of TC Dumile (10-min maximum sustained wind of 36 m.s$^{-1}$ or 70 kts at peak intensity) using the ORILAM (Organic Inorganic Log-normal Aerosol Model; Tulet, 2005) aerosol scheme and the LIMA (LIMA, Liquid Ice Multiple Aerosols; Vié et al., 2016) two-moment microphysics scheme. Using a one-moment microphysics scheme (ICE3, the scheme used in the present study) for the simulation of TC Dumile led to a larger, more symmetric and more intense system than using a two-moment microphysics scheme coupled with an aerosol scheme. It would be interesting to use the same aerosol-microphysics coupling for TC Enawo and





assess the impact on the model's representation of TTL ice and water vapor transport to the UTLS, however the
computation time is expected to increase by a factor 3. In addition, Jiang et al. (2019) investigated the effects of
sea salt aerosols on precipitation and UTLS water vapor in TCs with the WRF model. They simulated typhoon
Hato in the Northwestern Pacific using two nested domains with horizontal resolutions of 9 km and 3 km. They
performed two numerical experiments with the same two-moment microphysics scheme but different sea salt
emission intensity. The experiment with increased sea salt emission intensity indicated an increase in precipitation
as well as more intense vertical movement in the eyewall and thus more water vapor transport to the upper
troposphere, which promoted cloud ice deposition. This experiment was drier in the UTLS (above 17 km) and
had a 18-20 km domain-averaged water vapor mixing ratio ~0.02 lower than the experiment with reduced sea salt
emission. The enhanced air drying was explained by the enhancement of cloud ice deposition growth, which
consumes more water vapor in the upper troposphere. The aerosol-microphysics schemes presented in Hoarau et
al. (2018) can be used in an ocean-waves-atmosphere coupled system (Pianezze et al., 2018) to include the effects
of sea-salt aerosols on the microphysical structure of tropical cyclone. It could also be interesting to use this
version of Meso-NH to further investigate water vapor transport to the UTLS in TCs.

The large uncertainty in our estimate of net flux of water vapor requires additional modeling (convection-
permitting) and analysis of TTL in-situ/satellite observations of TCs in various basins but this is left to further
study. A preliminary analysis of 17-years MLS observations of water vapor for the most intense TCs (categories
4&5 on the Saffir-Simpson Scale) over 3 TC basins (Western North Pacific, North Atlantic and Soutwestern
Indian Ocean) show that these storms moisten the lower stratosphere (83 hPa/~18 km) from 0.3 to 0.5 ppmv in a
~5 to 10° region around the eye.

## 7. Summary and Conclusions

We extended the analysis of Evan et al. (2020) on the impact of TC Enawo on the TTL water vapor using the
Meso-NH cloud resolving Mesoscale model. Between 2 and 4 March, Enawo intensified from TD to TS. On 4
March, Enawo stalled over the ocean while intensifying. Enawo evolved from TS to TC on 5 March, and reached
peak intensity on 7 March.

The Meso-NH high-resolution (2 km) simulation used in this study covers Enawo's lifecycle from 2 to 7 March
for a model domain covering the SWIO. Five configurations were used to test the sensitivity of the model to
domain extent, horizontal resolution, and SST boundary conditions.

The westward propagation of Enawo was relatively well represented by the model simulations with comparable
propagation speed to the best track, and a maximum deviation between the simulated and observed tracks of
~100km. All the simulations were able to show that Enawo stalled over the ocean while intensifying on 4 March.
Overall, simulation S4 achieved the most realistic representation of Enawo's intensity and propagation.
Precipitations in simulation S4 were compared to GPM satellite observed rainfall. Precipitation in the eyewall
tends to be overestimated in simulation S4, while precipitation within the secondary rainbands were consistent
with those observed by GPM. Infrared brightness temperatures in simulation S4 were compared to the
METEOSAT 8 satellite images. The horizontal structure of the cloud tops within Enawo is relatively well
represented. Simulation S4 had slightly higher convective clouds, as simulated convective clouds were 2°C colder
than the observed ones.



Profiles of IWC in simulation S4 were compared through a vertical cross section of a CALIPSO overpass over the eyewall on 5 March 21:30 UTC. Observed and simulated IWC have similar ranges in the upper troposphere but simulated ice above the tropopause is overestimated by a factor 10. The location of the convective center was fairly well reproduced by the simulation, but the distribution ice above the CPT in simulation S4 covers a larger region than in CALIPSO.

Net mass fluxes of water vapor and ice into the stratosphere were estimated within a 700 km region around Enawo's center. Injection of water vapor and ice was maximum during the intensification phase of Enawo, when it stalled over the ocean on 4 March. Injection of ice decreased after March 5, although Enawo reached its peak intensity on 7 March.

An overshooting convection event on 4 March was analysed in detail. Direct injection of ice was simulated between 17 and 19 km altitude, associated with dehydration below the tropopause. Ice between 17 and 19 km altitude sublimated in a subsaturated environment and thereby hydrated the lower stratosphere. On average, the overshooting event simulated on 4 March 12UTC increased the lower stratospheric water vapor mixing ratio by 1.3 ppmv in a 50 km region around the TC center.

Water vapor profiles simulated by the 2 km Meso-NH simulations were averaged over a 500 km region surrounding Enawo's center. A negative water vapor change of -0.1 ppmv was found at 16.5 km altitude near the tropical tropopause. Between 17 and 19 km in altitude, an increase of ~2 ppmv in water vapor mixing ratio was simulated.

Comparison of net mass flux of water vapor and ice indicated that positive net mass fluxes of water vapor into the stratosphere are not necessarily associated with direct injection of ice by overshooting convection. The positive anomaly in the lower stratosphere started from the eye region of the storm and propagated outward Enawo's center. At the end of the simulation, the water vapor mixing ratio background rose by 1.5ppmv, an increase of 45%, 500 km away from Enawo's center.

Our simulation of TC Enawo suggests that TCs over the SWIO (0-30°S, 30-90°E) could produce a moistening of 0.4 ppmv over a region which represents ~8% of the global tropics (30°S-30°N). We extended our results to estimate the global impact of TCs to the tropical lower stratospheric water vapor using the estimates from published work (Allison et al., 2018; Preston et al., 2019) and by calculating statistics on TC numbers and durations using the IBTrACS dataset. We estimated a global impact of TC induced tropical lower stratospheric moistening of 0.3 to 0.5 ppmv.

*Data availability.* METEOSAT 8 data are accessible at https://disc.gsfc.nasa.gov/datasets/ 371 GPM MERGIR 1/summary. The GPM are provided by G. Huffman, D. Bolvin, D. Braithwaite, K. Hsu, R. Joyce, P. Xie, 2014: Integrated Multi-satellitE Retrievals for GPM (IMERG), version 4.4. NASA's Precipitation Processing Center, accessed 29 September, 2020, ftp://arthurhou.pps.eosdis.nasa.gov/gpmdata/. V4.10 CALIPSO Level 2 5 km cloud layer product is available at: https://doi.org/10.5067/CALIOP/CALIPSO/LID_L2_05KMCLAY-STANDARD-V4-10. The IBTrACS data are provided by Knapp, Kenneth R.; Diamond, Howard J.; Kossin, James P.; Kruk, Michael C.; Schreck, Carl J. III (2018). International Best Track Archive for Climate Stewardship (IBTrACS) Project, Version 4. NOAA National Centers for Environmental Information. https://doi.org/10.25921/82ty-9e16. The CFH water vapor data is available from the SE upon request.

*Author contributions.* All authors contributed to the paper. DH and SE wrote the manuscript with contributions from JP, TD, JB, KR, VN, SB, CB and JPC. DH performed the Meso-NH simulations. All authors revised the manuscript draft.

*Competing interests.* The authors declare that they have no conflict of interest.



*Acknowledgments*

620 This work was partly funded by the French LEFE CNRS-INSU Program (VAPEURDO) and by the French Agence Nationale de la Recherche CONCIRTO project (ANR375 17-CE01-0005-01). Computer resources were allocated by GENCI (project 1100297). The Meso-NH code is publicly available at http://www.mesonh.aero.obs-mip.fr/. The simulations were performed with version 5-3. This work was done during the PhD of D. Héron who was financially supported by a MENRT fellowship from the University of Réunion Island.

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



**Table 1: Definitions of Meso-NH simulations used in the study. All simulations are from 2 March 00UTC to 7 March 00 UTC just before Enawo's landfall over Madagascar. Root Mean Square Error (RMSE) between the Best Track and simulated values of track (in km) and MSLP (in hPa) are calculated for each simulation. The values in brackets represent the standard deviation.**

| Simulation name | Horizontal grid-spacing (km) | Model domain | Interactive SST | Initial and boundary conditions | Track RMSE (km) | MSLP RMSE (hPa) |
|---|---|---|---|---|---|---|
| S1 | 10 | A | Forced (IFS) | ECMWF IFS | 167±68 | 7.4±5.8 |
| S2 | 5 | A | Forced (IFS) | ECMWF IFS | 141±67 | 8.0±7.1 |
| S3 | 2 | A&B | Forced (IFS) | ECMWF IFS | 105±50 | 6.2±5.5 |
| S4 | 2 | B | Forced (IFS) | ECMWF IFS | 97±54 | 5.7±5.1 |
| S5 | 2 | B | Coupled (CROCO) | ECMWF IFS | 87±48 | 8.9±8.7 |


**Table 2: TC total net flux of water vapor (x $10^3$ t.hr$^{-1}$) to the lower stratosphere from Allison et al. (2018) and from the present study.**

| | Note | $F_{mean}$ (x $10^3$ t.hr$^{-1}$) |
|---|---|---|
| Allison et al. (2018) | 300-km Cat 1 TC Ingrid in the NA 1.33 km grid-spacing, net flux estimated over 43 hours | 4.6 |
| The present study | 600-km Cat 3 TC Enawo in the SWIO 2 km grid-spacing, net flux estimated over 48 hours | 2.7 |

**Table 3: The 25th, mean (standard deviation), and 75th percentiles for each region and global tropics of the annual number of tropical cyclones with 1-min VMAX ≥ 64 kt. The 25th, mean (standard deviation), and 75th percentiles for each region and global tropics of the duration of tropical cyclones (hours). The duration of a TC is defined as the period during which VMAX ≥ 64kt. The 25th, mean (standard deviation), and 75th percentiles for each region and global tropics of maximum 1-min sustained wind (kt) at peak intensity.**

| Region | Number of TC (1-min VMAX ≥ 64 kt) | | | Duration (hours) | | | 1-min VMAX (kt) at peak intensity | | |
|---|---|---|---|---|---|---|---|---|---|
| | 25th | Mean | 75th | 25th | Mean | 75th | 25th | Mean | 75th |
| Western Pacific | 12.2 | 14.5±4.4 | 17.0 | 72 | 88±18 | 100 | 91.6 | 95.8±5.7 | 99.6 |





| | | | | | | | | |
|---|---|---|---|---|---|---|---|---|
| Eastern Pacific | 7.0 | 9.3±3.6 | 11.0 | 64 | 78±20 | 92 | 90.5 | 96.0±9.5 | 102.9 |
| South Indian | 6.2 | 8.3±2.8 | 10.0 | 59 | 71±19 | 79 | 91.5 | 97.6±8.8 | 102.5 |
| North Atlantic | 4.0 | 6.4±3.1 | 8.7 | 32 | 51±25 | 70 | 84.0 | 92.6±9.4 | 98.7 |
| South Pacific | 4.0 | 5.5±2.5 | 7.0 | 56 | 73±30 | 82 | 85.2 | 96.8±12.5 | 106.8 |
| North Indian | 1.0 | 1.0±1.0 | 2.0 | 30 | 48±24 | 70 | 81.2 | 94.0±17.2 | 105.0 |
| **Global** | **42.5** | **45.0±9.1** | **49.0** | **102.0** | **78±19** | **186.0** | **79.5** | **96.2±21.4** | **113.6** |


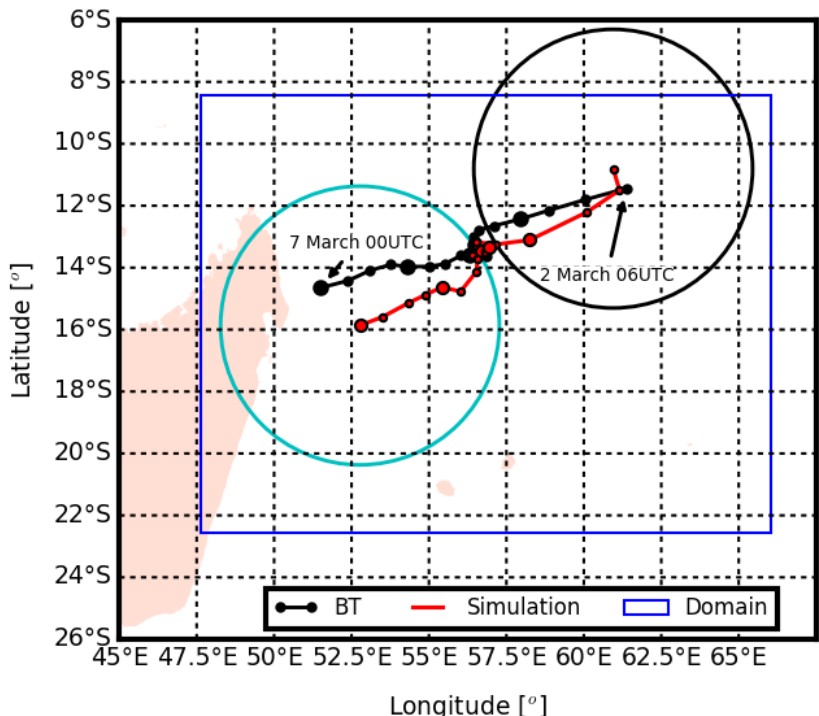

**Figure 1: Simulated (red line) versus RSMC La Réunion best-track (black line). Small dots correspond to 6-hourly**
**locations while large dots correspond to daily locations from 2 March 06UTC to 7 March 00UTC. The 2 circles indicate**
**regions which are used to compute an average water vapor profile around the TC center for the beginning and the end**
**of simulation S4. The domain used in the Meso-NH simulation is in blue and has a 2-km resolution.**



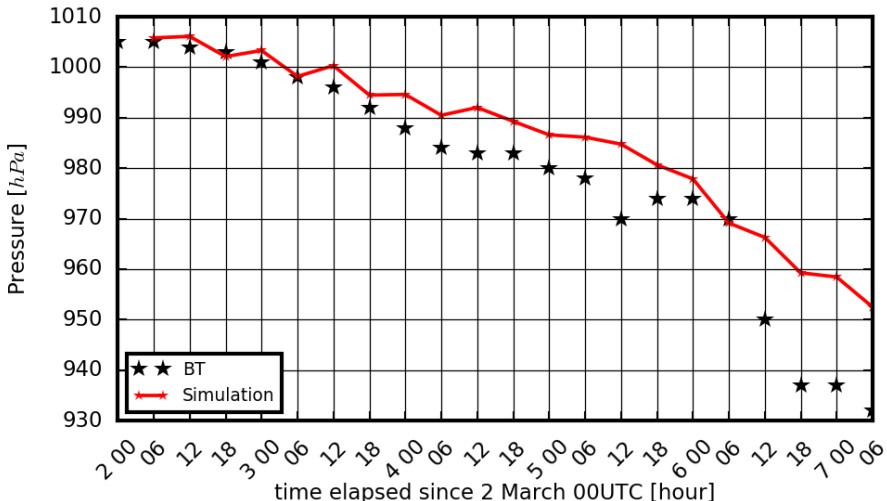

**Figure 2: Time series of MSLP in the Meso-NH 2km simulation (red line) and in the RSMC La Réunion best-track data (black stars). 6-hourly values of MSLP are shown from 2 March 00UTC to 7 March 00UTC.**

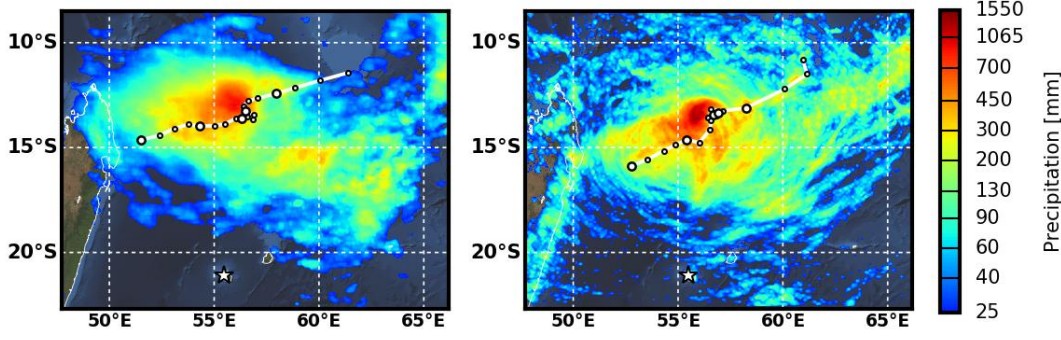

**Figure 3: Accumulated precipitation for the period 2 March 06 UTC to 7 March 00UTC. Left: GPM observation. Right: Simulated precipitation. The white line on the left and right panels corresponds to RSMC La Réunion best-track data and simulated best-track respectively.**


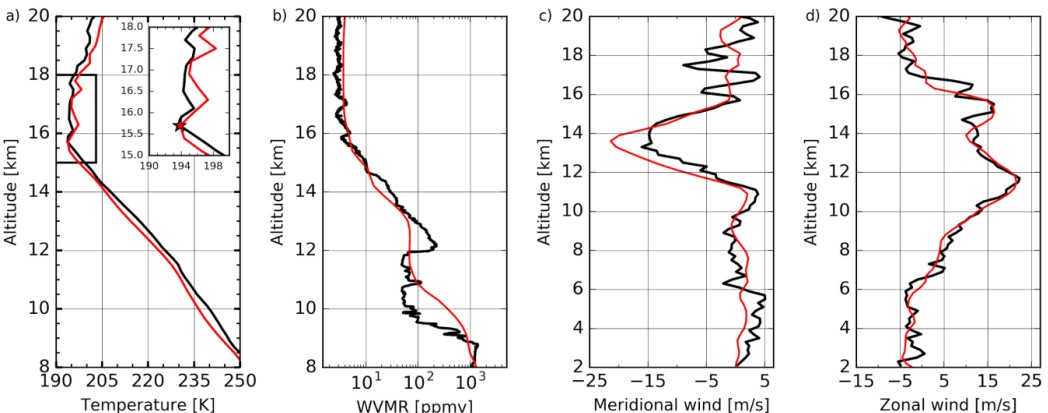

**Figure 4: Simulated (red) versus observed (black) profiles at the location of the Maïdo Observatory on 3 March 18 UTC: a) temperature, b) water vapor mixing ratio, c) meridional wind and d) zonal wind. CFH and M10 sonde observations are shown in black while simulated fields are shown in red. On the temperature profiles the location of the observed/simulated Cold Point Tropopause is indicated by black and red stars respectively.**

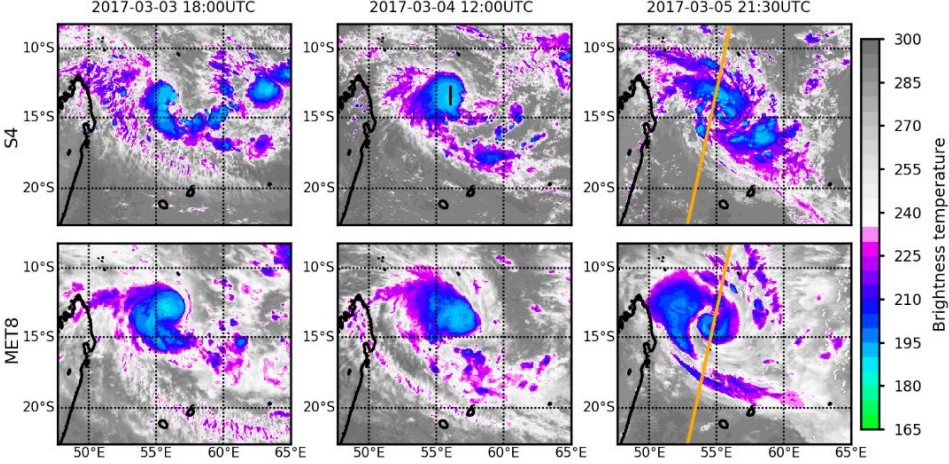

**Figure 5: Observed (bottom panels) versus simulated (upper panels) infrared brightness temperature on 3 March, 18 UTC (left), 4 March, 12 UTC (middle) and on 5 May, 21:30 UTC. The orange lines correspond to CALIPSO orbit tracks on 5 March at 21:30 UTC. On 4 March, 12 UTC the small vertical black line on the simulated infrared brightness temperature indicates the south-north cross section shown on Figure 8.**

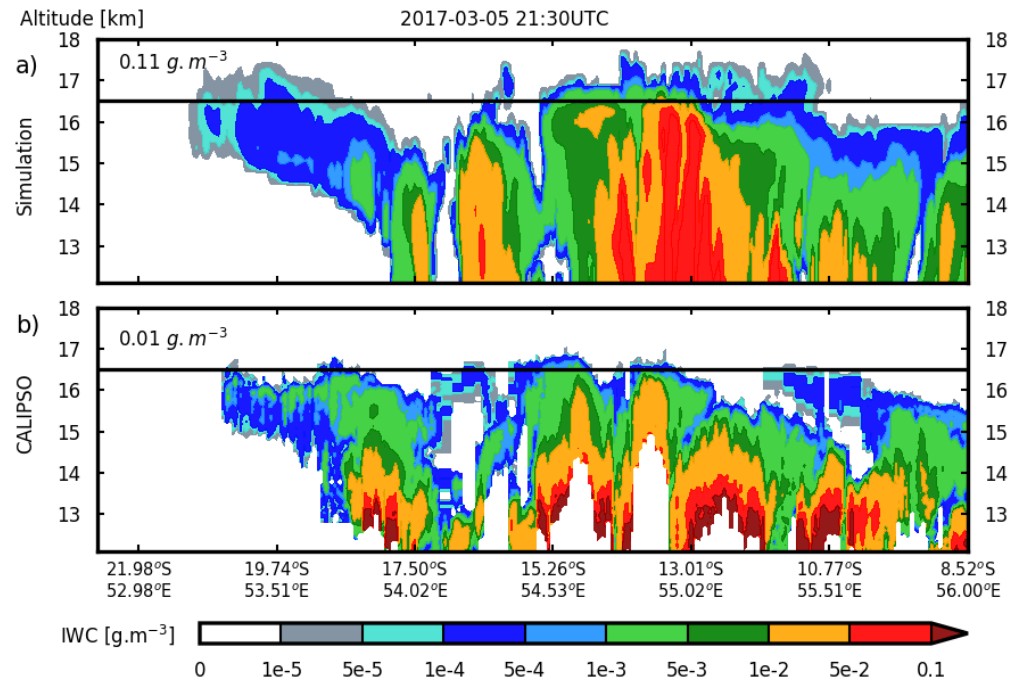


**Figure 6: Cross-section (Latitude versus altitude) of CALIOP Ice Water Content (IWC in g m-3, bottom) along CALIOP track over TC Enawo on 5 March, 2130UTC. The corresponding simulated IWC for S4 is shown on the top panel. The black curves on both panels correspond to the simulated CPT which has a mean height of 16.5km. The**

**numbers on the top left corner of each plot indicate the mean IWC above the CPT.**




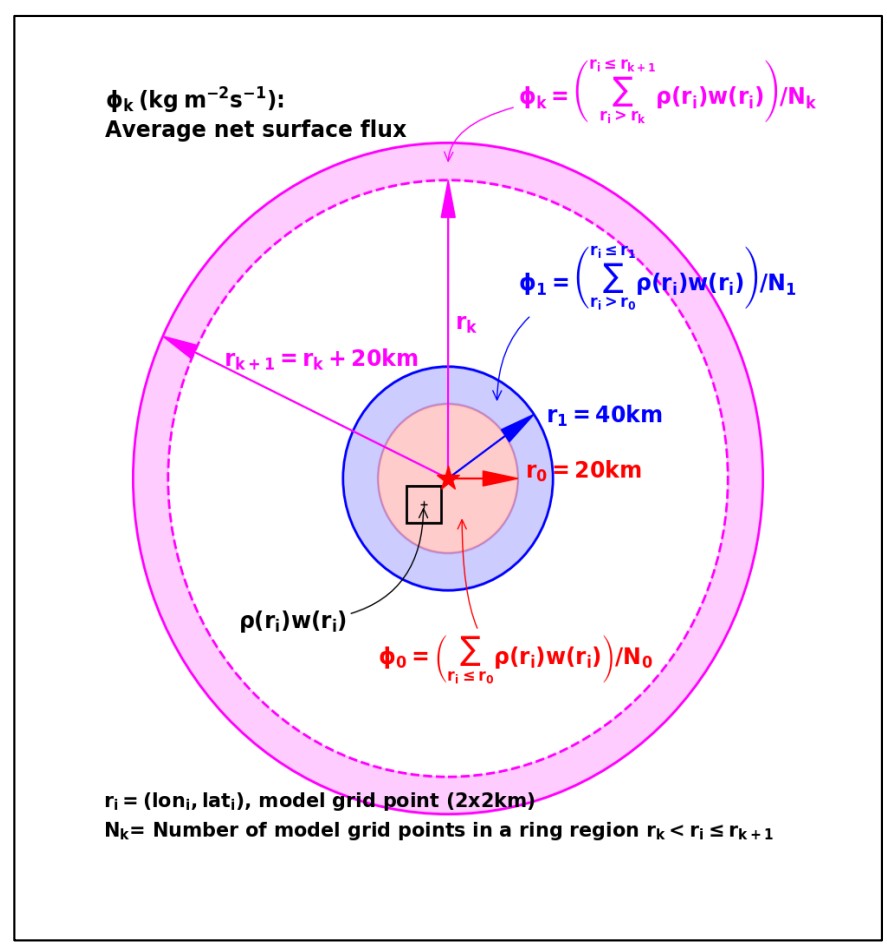

**Figure 7:** Diagram illustrating the computation of average net mass flux density (kg.m⁻².s⁻¹) in the simulation. Average net mass fluxes density is computed for ring regions around the TC center (red star) for 20 km radius increment.


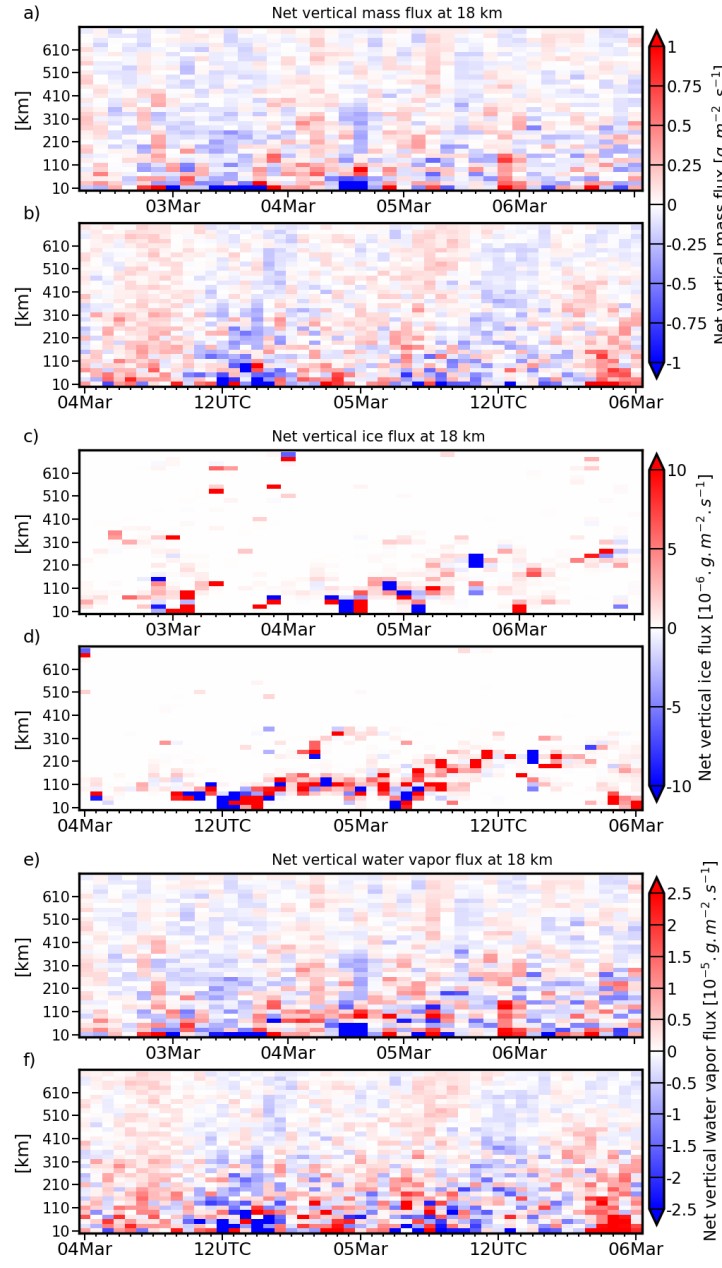

**Figure 8: a) Net mass flux at 18 km in the simulation as a function of time and distance from the TC center. Values were averaged over bins of 20 km x 20 km for a distance of 0 to 700 km from the TC center and are shown every 3 hours from 2 March, 06 UTC to 7 March, 00 UTC.  b)  same as a) but hourly-values from 4 March, 00 UTC to 6 March, 00 UTC while the storm was intensifying over the ocean. c) 3-hourly ice flux values at 18 km in the simulations from 2 March, 06 UTC to 7 March, 00UTC d) Same as c) but for hourly values from 4 March, 00 UTC to 6 March 6, 00UTC. e) 3-hourly water vapor flux values at 18 km in the simulations from 2 March, 06 UTC to 7 March, 00 UTC d) Same as c) but for hourly values from 4 March, 00 UTC to 6 March 6, 00 UTC.**





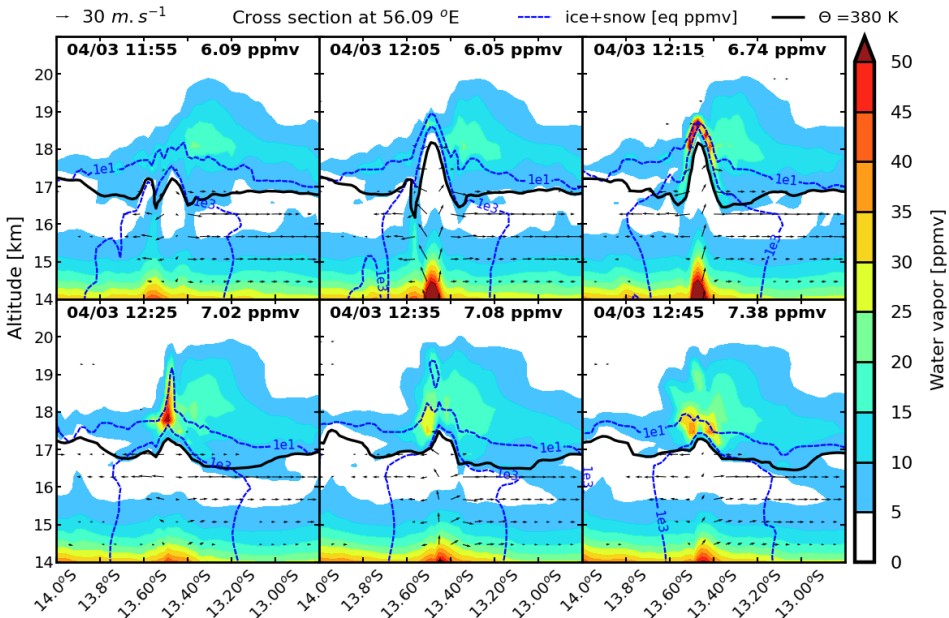

**Figure 9: North to south cross sections through the simulated overshooting cloud on 4 March. Plots are every 10 minutes on 4 March from 11:55 UTC to 12:45 UTC. Color contours correspond to the water vapor mixing ratio. Ice mixing ratio (Snow+cloud ice) values ranging from 1 to 1000 ppmv are indicated by blue dashed lines. Black arrows correspond to the zonal/vertical wind fields. The black line in all panels corresponds to the 380 K isentropic surface. Mean water vapor mixing ratios above the 380 K surface are indicated in black on the top right corner of each panel.**







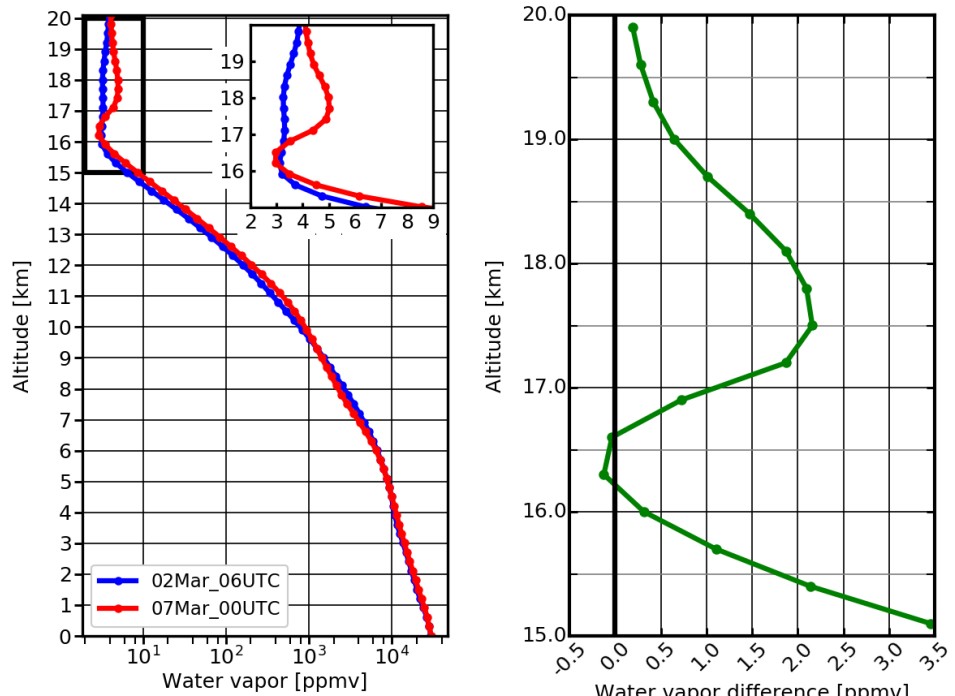

**Figure 10: Left, simulated average water vapor mixing ratio profiles on 2 March 06 UTC (blue line) and 7**
**March 00 UTC (red line). The profiles were averaged over a 500 km region surrounding Enawo's center**
**which is indicated by the circles on Figure 1. Right, water vapor mixing ratio difference between 2 March**
**06 UTC and 7 March 00 UTC. The profiles were averaged over a 500 km region surrounding Enawo's**
**center which is indicated by the circles on Figure 1.**




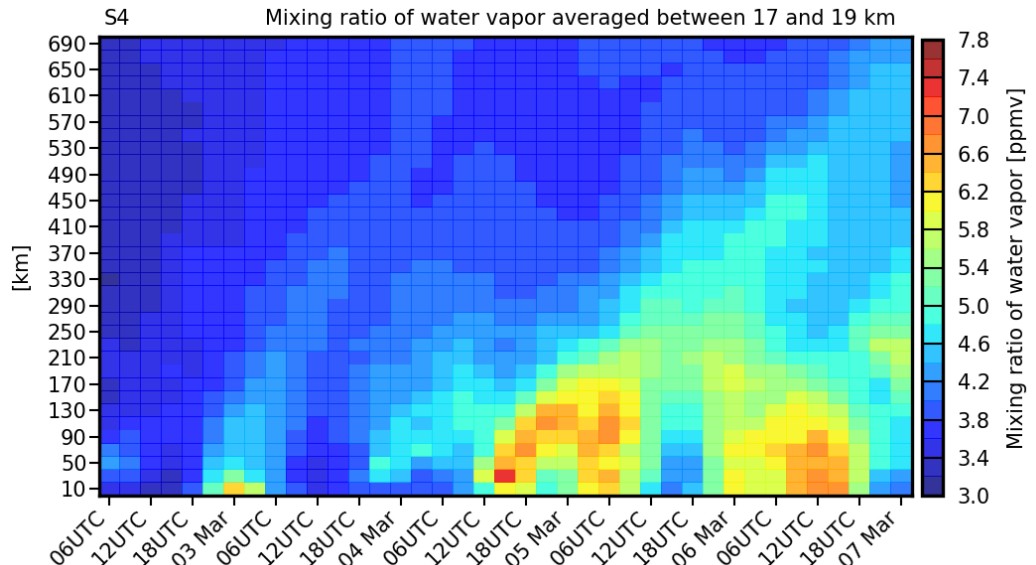

**Figure 11: Time versus distance from the TC center of simulated water vapor mixing ratio averaged over the 17-19 km altitude range. Values correspond to bins of 20 km for a radius of 0 to 700 km around the TC center and are shown every 3 hours from 2 March 06 UTC to 7 March 00 UTC.**


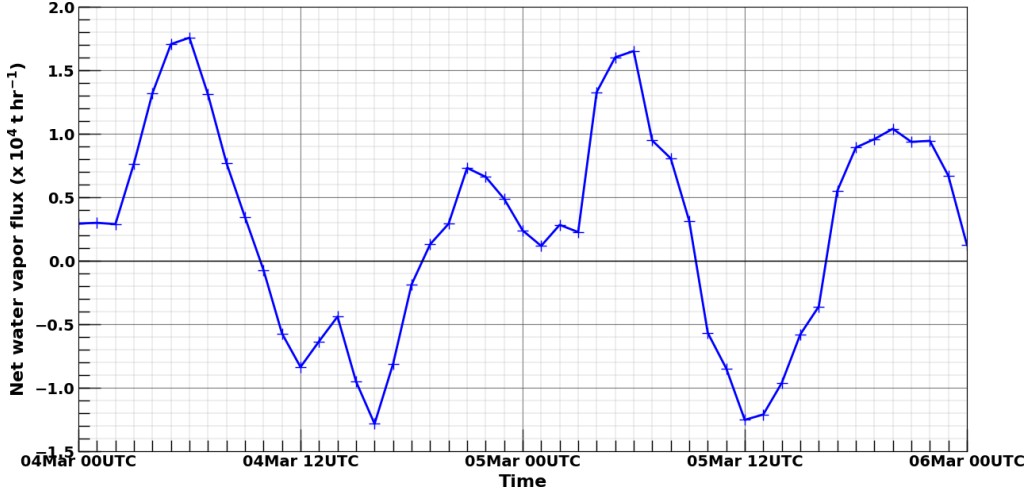

**Figure 12: Time evolution of net vertical water vapor flux at 18 km (t.hr⁻¹) from 4 March 00 UTC to 6 March 00 UTC. A 700-km region surrounding Enawo's center was used to compute the net vertical water vapor flux for each hourly model output.**
