# Peer review of "Mesoscale simulations of tropical cyclone Enawo (March 2017) and its impact on TTL water vapor."

_Atmospheric Chemistry and Physics, 2020_

## Referee Comment (RC1) · Anonymous Referee #1 · 18 Nov 2020

**Review of "Mesoscale simulations of tropical cyclone Enawo (March 2017) and its impact on TTL water vapor" by D. Heron et al.**

This paper is clear and well written. It provides a credible description of a convection-permitting model simulation of a tropical cyclone, including the overall structure, evolution, and precipitation. However, the focus of the paper is on quantifying the impact of TCs on stratospheric humidity. The comparison with CALIOP measurements shows that the simulated cloud has a factor of 10 too much ice above the tropopause. I cannot see the point in proceeding with using the simulation to quantify the humidification of the lower stratosphere; the estimate would obviously be excessive and unrealistic. Further, the extrapolation of results from a single, unrealistic simulation to the Southwest Indian Ocean (and even to the global tropics) is dubious. The resulting estimate has

little quantitative value. The authors state that Enawo was not even representative of TCs in the SWIO region (it was apparently the strongest TC in the summer hemisphere for the 2016/2017 season). I fear readers will simply take the stratospheric hydration numbers (0.3–0.5 ppmv) as realistic estimates when they clearly are not. As a result, I do not think the paper should be published in ACP.

---

## Referee Comment (RC2) · Anonymous Referee #2 · 1 Dec 2020

The article by Damien Héron and coauthors investigates the impact of deep convection associated with tropical cyclone Enawo in March 2017 on water vapour in the tropical tropopause layer using high-resolution cloud-resolving modeling with Meso-NH model. The effect of tropical cyclones on the gaseous composition of the tropical lower stratosphere is a topical issue given the lack of consistency across the various observational and/or modeling studies on this subject that provide highly variable estimates on the magnitude and even the sign of the effect of overshooting convection on TTL/LS water vapour.

The paper is fluently written and easy to follow. The experimental and modeling setup is comprehensively described. The results of MesoNH simulation are compared with various types of observations regarding the accumulated precipitation (GPM); water

vapour and state parameters (CFH balloon sounding from La Reunion island several hundred kilometers away from Enawo); IR brightness temperature (METEOSAT8) and ice water content derived from a CALIOP transact across the cyclone. The simulations are further reported as mass fluxes and vertical profiles of vapour and ice as well as the horizontal evolution of water vapour immediately above the tropopause relative to the TC center. The simulated cross-sections through the overshooting cloud suggest peak values of water vapour reaching 40-50 ppmv above 380 K level, whereas an average profile within 500 km radius shows an enhancement of up to 2.1 ppmv between 17 – 19 km, which is attributed to TTL hydration as a result of sublimated ice crystals detrained by overshooting updrafts of TC Enawo.

The results of simulation are upscaled to the global tropics using IBTrACS statistics on TCs and the authors conclude that the tropical cyclones are responsible for the global tropical lower stratosphere moistening of 0.3 – 0.5 ppmv. This is the central message of this study however as explained in the following, it suffers from very low credibility of Meso-NH simulations of this particular case.

The simulations are far from being realistic, they reveal important differences with observations regarding the key parameters: temperature, water vapour and ice water. In particular, the simulated temperature profile (Fig. 4a) doesn't agree with the radiosounding neither in vertical structure nor in the absolute values showing differences reaching 2 K, which translates to RHi bias of up to about 50%. The simulated water vapour shows a flat profile above CPT with a wet bias of about 1 ppmv compared to the measurements. The warm and wet bias suggests that the model does not properly account for dehydration process.

A more important issue is that the simulation overestimates the IWC by a factor of 10 and shows significant amount of condensed water well above the CPT, where CALIOP does not see any ice crystals (Fig. 6). This renders further estimates of moistening due to ice sublimation highly uncertain. While a certain effort towards evaluation of simulations against the observations is made within this study, it is far from satisfactory as far

as the key parameters are concerned. In particular, the IWC validation could benefit from more than one CALIOP samplings of Enawo within the simulated domain as well as from similar measurements by CATS lidar. The water vapour could be compared to the respective MLS/SAGEIII/ACE-FTS observations, which could also serve to assess the bulk effect of Enawo TC on the TTL/LS water vapour. Finally, the simulated thermal structure, which is crucial for the moistening potential of detrained ice, could be compared with GPS-RO temperature profiling or with ERA reanalysis assimilating these measurements with high level of confidence. None of this has been done for this study.

The unrealistic nature of mesoscale simulations together with a simplified approach to their upscaling and unclear representability of TC Enawo renders the central conclusion of the manuscript misleading. Hence, I cannot recommend it for publication at ACP.

---

## Author Comment (AC1) · 8 Feb 2021

**Answers to Referees # 1 and # 2.**

We would like to thank the referees for their time and effort in evaluating our paper. We are aware that the differences in IWC in the stratosphere between Meso-NH and CALIOP seem to be excessive. The temperature difference of up to 2K in the TTL between the simulation and the radiosonde observation is also an issue. However, this is a stringent test as we compare the model to a single observation location. The Meso-NH model does a good job of reproducing the tropical tropopause height at 16.5 km, which suggests that the 300-m vertical resolution is adequate to represent the sharp tropopause. Additional comparison with COSMIC GPS profile or ERA 5 reanalysis could be useful and this will be done. It is not clear if the temperature bias is due to the representation of radiative processes in the model. Gettelman et al. (2004) examined the radiation balance of the TTL and showed that the temperature in the TTL is influenced in part by the ozone distribution. Therefore, it is important to accurately capture changes in ozone in the TTL, as it will strongly influence the net radiative heating of this region. Evan et al. (2013) showed that using a more realistic ozone climatology could improve the representation of TTL temperature in a mesoscale model. This could be a way to improve the representation of TTL temperature in the Meso-NH simulation of TC Enawo.

As for the apparent wet bias above 16 km on Figure 4, we would like to emphasize that CFH instrument launched on 3 March 2017 had a dry bias of 1 ppmv in the stratosphere but did perform well up to the tropopause. This issue was described in section 5.1 of Evan et al. (2020). The CFH sonde launched on Reunion island are not recovered after each flight as they land most of the time in the ocean, and thus it was not possible to examine in more details the CFH instrument after the flight on 3 March 2017. Thus it was hard to conclude whether the dry bias in the stratosphere was due to an instrument issue.

We performed numerous comparisons with observations to assess whether simulated TC Enawo's track, intensity, and structure were realistic. We concluded that TC Enawo simulation is satisfactory in terms of track (mean error of 100 km) and intensity (average difference of 5.7 hPa for the central pressure). Using different configurations of the WRF model for the simulation of TC Ingrid (i.e. different microphysics and cumulus parameterizations) and analysis nudging for the first 66 hours of simulations, Allison et al. (2018) found root mean square (RMS) errors of 1.30° (~143km) and 5.26 hPa for the track and central pressure respectively in their best simulation. We therefore concluded that our simulation of TC Enawo in terms of track, intensity and structure was as realistic as Allison et al. (2018). Our comparison of simulated ice in the TTL to CALIOP observation is not as good as the simulated track, intensity and structure. However we chose to show it in our article as we thought it was important to present the limitations of our model in an honest way. We think that considering our model results to be "obviously excessive and unrealistic" by considering only the CALIOP vertical cross section to be overly simplistic. Simulating TCs and their effects on the TTL at mesoscale is not an easy task and this study builds upon the 3-year PhD work of the first author. To our knowledge, neither the simulated TTL water vapor nor ice were validated in previous modeling studies of TC effects on the TTL water vapor.

We think that our estimate of the hydration of the TTL by TC Enawo in the Southern Indian Ocean (SIO) could still be valuable to the scientific community and complement the results found in Allison et al. (2018) for the following reasons:

- Our mesoscale simulation does not use any nudging, to let convective clouds develop in an unperturbed way. Hence, one should not expect a perfect match between the model results and the observations.

- We agree that the simulation seems to overestimate the vertical extent of ice clouds in the TTL above TC Enawo. Ongoing modeling work by Reinares Martinez et al. (2020, manuscript in preparation) shows that using a two-moment microphysics scheme (LIMA, Liquid Ice Multiple Aerosols; Vié et al., 2016) could improve the model results in terms of TTL ice clouds one compared to a one moment microphysics scheme (ICE3) as used in this study. However this is for simulating in-situ TTL cirrus formation (Reinares Martinez et al., 2020) and not for convectively detrained ice crystals. We did consider using the 2-moment LIMA microphysics scheme to simulate TC Enawo. Hoarau et al. (2018) performed a 2-km cloud resolving Meso-NH simulation of TC Dumile using the ORILAM (Organic Inorganic Log-normal Aerosol Model; Tulet, 2005) aerosol scheme and the LIMA two-moment microphysics scheme. Using a one-moment microphysics scheme (ICE3, the scheme used in the present study) for the simulation of TC Dumile led to a larger, more symmetric and more intense system than using a two-moment microphysics scheme coupled with an aerosol scheme. The computation time with LIMA is expected to increase by a factor 3, especially when running the model with such a high vertical resolution in the TTL/high model top as compared to more "traditional" simulation of TC. Therefore due to limited computational resources, we chose to do our analysis of TC Enawo's effects on the TTL with the ICE-3 one-moment microphysics scheme. In ICE-3, an implicit adjustment of the temperature, vapor, cloud, and ice contents is performed in clouds with a strict saturation criterion (Lac et al., 2018). This is a limitation of the scheme as supersaturation of up to 50% (RHice 120-150%) can be observed in the TTL (e.g. Jensen et al., 2013; Krämer et al., 2020). This will have an effect on the amount of ice crystals formed and could explain the higher IWC when compared to CALIOP. We plan to adjust the saturation threshold for ice formation in the TTL in future Meso-NH simulations of TC Enawo.

- There are very few in-situ measurements of IWC in the TTL over the SIO so to evaluate models' performance in simulating TTL ice above TC Enawo we use CALIOP observations. Specifically we use CALIOP v4 IWC values which are derived from retrievals of optical extinction. CALIOP V4 IWC parameterization does compare well with NOAA total water measurements during the ATTREX mission reported in Thornberry et al. (2017). However there can be large uncertainties for low values of IWC ($< 10^{-4}$ g/m$^3$). Deriving IWC from attenuated backscatter measurements is hampered by the complexity of global particle size distributions and particle habits, which determine the conversion from attenuated backscatter, which is a kind of ice particle cross-sectional area measurement, and ice water content, which requires knowing the volume of ice in a sample volume of air. Thus the factor of 10

between the simulated and "observed" IWC in the TTL is most likely an upper limit. We should have used model diagnostics similar to the actual observations, i.e backscatter to have a quantity directly comparable to the data measured by the CALIOP instrument. This will be further investigated. Nevertheless, we do agree that the vertical distribution of IWC on Figure 6 suggests that ice clouds are too high in the simulations. We have to further investigate whether ice is overestimated in the simulation because ice crystals generated in the TTL are too small/numerous and thus do not sediment or whether simulated convective updrafts are too important and result in too much ice being transported to the TTL.

- The average simulated water vapor anomaly on Figure 10 has a local minimum near the tropopause ~ 16.5 km and a local maximum of 2 ppmv at 17.5km. The profile of water vapor anomaly simulated for TC Enawo agrees in magnitude with water vapor anomalies simulated for TC Ingrid in the North Atlantic (cf Figure 9b of Allison et al., 2018), despite the fact that TC Ingrid was a category 1 hurricane and Enawo was a category 3 cyclone. We computed a water vapor difference profile for MLS profiles close to the TC center on March 4, 00UTC and March 6, 00UTC. MLS water vapor profiles within ±5 ∘ latitude and ±10∘ longitude of the TC center on those dates were selected for a period of +- 3 hours surrounding March 4, 00 UTC and March 6, 00 UTC. 5 and 6 coincident MLS profiles were found respectively on March 4 and March 6. The mean of these water profiles and their difference in the TTL is shown below.

[Figure]

Moistening of +0.3ppmv can be seen at 82hPa with drying at the tropopause in agreement with the simulated water vapor difference in the vicinity of the TC Center. The broad MLS averaging kernel (~3 km) may limit the ability to detect narrow layers with high humidity. To compare the high-resolution simulated water vapor profile to the MLS satellite data, we should smooth the high-resolution simulated water vapor profiles to match the resolution of the satellite profiles using the MLS vertical averaging kernels.

Recently, in situ TTL measurements presented in Jensen et al. (2020) showed that direct injection of water vapor by convection can reach 67hPa or ~18.5 km in altitude, similar to the

vertical injection of overshooting convection in our simulation. In Jensen et al. (2020), a positive anomaly of ~1.5 ppmv at 17.5km was related to a convective hydration event over Typhoon Haima, 500 m above the local tropopause.

- Finally, we conducted a preliminary analysis based on MLS water vapor measurements at 82 hPa to estimate an average water vapor anomaly due to TCs in the SWIO following methodology of Ray and Rosenlof (2007.) We considered 129 Tropical Storms/Tropical Cyclones in the SIO over the period 2004-2018 and computed averages centered on TCs from 2004 to 2018 of MLS water vapor anomaly at 82 hPa. The result is shown below.

[Figure]

The x and y axis are degrees in longitude and latitude relative to the center of TCs. We found an average anomaly of 0.2 to 0.3 ppmv around a 5°x5° region (roughly 500x500km). This preliminary analysis suggests that TCs in the SIO can hydrate the lower stratosphere in the vicinity of the storm. This is coherent with the positive water vapor anomaly simulated at 18.5 km in Meso-NH although with a weaker amplitude. We do recognize that our upscaling of the model results to the global tropics using IBTrACS statistics on TCs may not be realistic and overestimated.

Ongoing work with MLS data to apply the methodology of Ray and Rosenlof (2007) to assess hydration of the UTLS by tropical cyclones for the period 2004–2018 around the globe is under way. In addition, we plan to use the ERA 5 reanalysis temperature data, cloud observations from MODIS and Lagrangian trajectories to assess the convective origin of possible high water vapor measurements in the lower stratosphere above TCs. This will be presented in a future study.

Revised Meso-NH simulations of TC Enawo and its effect on TTL water vapor will be presented in a separate study.

**References:**

Allison, T., Fuelberg, H., and Heath, N.: Simulations of vertical water vapor transport for TC Ingrid (2013), J. Geophys. Res., 123, 8255–8282, https://doi.org/10.1029/2018JD028334, 2018.

Evan, S., Rosenlof, K. H., Dudhia, J., Hassler, B., and Davis, S. M. (2013), The representation of the TTL in a tropical channel version of the WRF model, *J. Geophys. Res. Atmos.*, 118, 2835–2848, doi:10.1002/jgrd.50288.

Evan, S., Brioude, J., Rosenlof, K., Davis, S. M., Vömel, H., Héron, D., Posny, F., Metzger, J.-M., Duflot, V., Payen, G., Vérèmes, H., Keckhut, P., and Cammas, J.-P.: Effect of deep convection on the tropical tropopause layer composition over the southwest Indian Ocean during austral summer, Atmos. Chem. Phys., 20, 10565–10586, https://doi.org/10.5194/acp-20-10565-2020, 2020.

Gettelman, A., P. M. d. F. Forster, M. Fujiwara, Q. Fu, H. Vömel, L. K. Gohar, C. Johanson, and M. Ammerman (2004), Radiation balance of the tropical tropopause layer, *J. Geophys. Res.*, 109, D07103, doi: 10.1029/2003JD004190.

Jensen, E. J., Pan, L. L., Honomichl, S., Diskin, G. S., Krämer, M., & Spelten, N., et al. (2020). Assessment of observational evidence for direct convective hydration of the lower stratosphere. *Journal of Geophysical Research: Atmospheres*, 125, e2020JD032793. https://doi.org/10.1029/2020JD032793

Krämer, M., Rolf, C., Spelten, N., Afchine, A., Fahey, D., Jensen, E., Khaykin, S., Kuhn, T., Lawson, P., Lykov, A., Pan, L. L., Riese, M., Rollins, A., Stroh, F., Thornberry, T., Wolf, V., Woods, S., Spichtinger, P., Quaas, J., and Sourdeval, O.: A microphysics guide to cirrus – Part 2: Climatologies of clouds and humidity from observations, Atmos. Chem. Phys., 20, 12569–12608, https://doi.org/10.5194/acp-20-12569-2020, 2020.

Lac, C., Chaboureau, J.-P., Masson, V., Pinty, J.-P., Tulet, P., Escobar, J., Leriche, M., Barthe, C., Aouizerats, B., Augros, C., Aumond, P., Auguste, F., Bechtold, P., Berthet, S., Bielli, S., Bosseur, F., Caumont, O., Cohard, J.-M., Colin, J., Couvreux, F., Cuxart, J., Delautier, G., Dauhut, T., Ducrocq, V., Filippi, J.-B., Gazen, D., Geoffroy, O., Gheusi, F., Honnert, R., Lafore, J.-P., Lebeaupin Brossier, C., Libois, Q., Lunet, T., Mari, C., Maric, T., Mascart, P., Mogé, M., Molinié, G., Nuissier, O., Pantillon, F., Peyrillé, P., Pergaud, J., Perraud, E., Pianezze, J., Redelsperger, J.-L., Ricard, D., Richard, E., Riette, S., Rodier, Q., Schoetter, R., Seyfried, L., Stein, J., Suhre, K., Taufour, M., Thouron, O., Turner, S., Verrelle, A., Vié, B., Visentin, F.,

Vionnet, V., and Wautelet, P.: Overview of the Meso-NH model version 5.4 and its applications, Geosci. Model Dev., 11, 1929–1969, https://doi.org/10.5194/gmd-11-1929-2018, 2018.

Ray, E. A. and Rosenlof, K. H.: Hydration of the upper troposphere by tropical cyclones, J. Geophys. Res., 112, D12311, https://doi.org/10.1029/2006JD008009, 2007

Reinares Martínez, I., Evan, S., Wienhold, F. G., Brioude, J., Jensen, E. J., Thornberry, T. D., et (2020). Unprecedented observations of a nascent in situ cirrus in the tropical tropopause layer. Geophysical Research Letters, 47, 2020GL090936. https://doi.org/10.1029/2020GL090936

Thornberry, T. D., Rollins, A. W., Avery, M. A., Woods, S., Lawson, R. P., Bui, T. V., and Gao, R.‑S. (2017), Ice water content‑extinction relationships and effective diameter for TTL cirrus derived from in situ measurements during ATTREX 2014, J. Geophys. Res. Atmos., 122, 4494– 4507, doi:10.1002/2016JD025948.